# Dynamics of human protein kinase Aurora A linked to drug selectivity

**Warintra Pitsawong[1†], Vanessa Buosi[1†], Renee Otten[1†], Roman V Agafonov[1†], Adelajda Zorba[1], Nadja Kern[1], Steffen Kutter[1], Gunther Kern[1], Ricardo AP Pádua[1], Xavier Meniche[2], Dorothee Kern[1]***

[1]Department of Biochemistry, Howard Hughes Medical Institute, Brandeis University, Waltham, United States; [2]Department of Microbiology and Physiological Systems, University of Massachusetts Medical School, Worcester, United States

**Abstract** Protein kinases are major drug targets, but the development of highly-selective inhibitors has been challenging due to the similarity of their active sites. The observation of distinct structural states of the fully-conserved Asp-Phe-Gly (DFG) loop has put the concept of conformational selection for the DFG-state at the center of kinase drug discovery. Recently, it was shown that Gleevec selectivity for the Tyr-kinase Abl was instead rooted in conformational changes after drug binding. Here, we investigate whether protein dynamics after binding is a more general paradigm for drug selectivity by characterizing the binding of several approved drugs to the Ser/Thr-kinase Aurora A. Using a combination of biophysical techniques, we propose a universal drug-binding mechanism, that rationalizes selectivity, affinity and long on-target residence time for kinase inhibitors. These new concepts, where protein dynamics in the drug-bound state plays the crucial role, can be applied to inhibitor design of targets outside the kinome.

DOI: https://doi.org/10.7554/eLife.36656.001

**\*For correspondence:**
dkern@brandeis.edu

[†]These authors contributed equally to this work

**Competing interests:** The authors declare that no competing interests exist.

## Introduction

Protein kinases have become the number one drug target of the 21th century (*Cohen, 2002*; *Hopkins and Groom, 2002*), due to their central role in cellular processes and involvement in various types of cancer (*Carvajal et al., 2006*; *Gautschi et al., 2008*; *Katayama and Sen, 2010*). Despite their therapeutic significance, the development of specific kinase inhibitors proves to be extremely challenging because they must discriminate between the very similar active sites of a large number of kinases in human cells. One of the biggest success stories is Gleevec: a highly selective drug that specifically targets Abl kinase, providing an efficient treatment of chronic myelogenous leukemia (CML) and minimizing side effects (*Iqbal and Iqbal, 2014*). Despite being a multi-billion-dollar cancer drug, the mechanism responsible for its impressive selectivity has been elusive until recently. Seminal work by the Kuriyan lab demonstrated that Gleevec can only bind to an inactive DFG (for Asp-Phe-Gly) loop conformation in the 'out-conformation' due to steric clash of the active, DFG-*in* conformation (*Nagar et al., 2002*; *Schindler et al., 2000*; *Seeliger et al., 2007*). Since then it has long been proposed that the conformational state of the fully-conserved DFG loop (*Taylor et al., 2012*) dictates the selectivity for Gleevec and other kinase inhibitors (*Lovera et al., 2012*; *Nagar et al., 2002*; *Schindler et al., 2000*; *Treiber and Shah, 2013*; *Xu et al., 1997*). The orientation of the DFG-motif and its possible steric clashes is indeed important for the ability of a class of inhibitors to bind to the kinase, but proved insufficient to explain drug selectivity and affinity. Earlier elegant work on Src and Abl recognized this and explored other hypotheses (e.g., differences in drug-binding pocket, energetic changes remote from the binding site and a conformational-selection mechanism) to reconcile the differences in Gleevec binding (*Dar et al., 2008*; *Levinson et al., 2006*; *Seeliger et al., 2007*; *2009*), but without conclusive success. Recent quantitative binding kinetics combined with ancestral

**eLife digest** Protein kinases are a family of enzymes found in all living organisms. These enzymes help to control many biological processes, including cell division. When particular protein kinases do not work correctly, cells may start to divide uncontrollably, which can lead to cancer. One example is the kinase Aurora A, which is over-active in many common human cancers. As a result, researchers are currently trying to design drugs that reduce the activity of Aurora A in the hope that these could form new anticancer treatments.

In general, drugs are designed to be as specific in their action as possible to reduce the risk of harmful side effects to the patient. Designing a drug that affects a single protein kinase, however, is difficult because there are hundreds of different kinases in the body, all with similar structures. Because drugs often work by binding to specific structural features, a drug that targets one protein kinase can often alter the activity of a large number of others too.

Gleevec is a successful anti-leukemia drug that specifically works on one target kinase, producing minimal side effects. It was recently discovered that the drug works through a phenomenon called 'induced fit'. This means that after the drug binds it causes a change in the enzyme's overall shape that alters the activity of the enzyme. The shape change is complex, and so even small structural differences can change the effect of a particular drug.

Do other drugs that target other protein kinases also produce induced fit effects? To find out, Pitsawong, Buosi, Otten, Agafonov et al. studied how three anti-cancer drugs interact with Aurora A: two drugs specifically designed to switch off Aurora A, and Gleevec (which does not target Aurora A).

The two drugs that specifically target Aurora A were thought to work by targeting one structural feature of the enzyme. However, the biochemical and biophysical experiments performed by Pitsawong et al. revealed that these drugs instead work through an induced fit effect. By contrast, Gleevec did not trigger an induced fit on Aurora A and so bound less tightly to it.

In light of these results, Pitsawong et al. suggest that future efforts to design drugs that target protein kinases should focus on exploiting the induced fit process. This will require more research into the structure of particular kinases.

DOI: https://doi.org/10.7554/eLife.36656.002

sequence reconstruction put forward a mechanism where an induced-fit step after drug binding is the key determinant for Gleevec's selectivity (*Agafonov et al., 2014*; *Wilson et al., 2015*), and fully recapitulates the binding affinities.

Here we ask the question whether this fundamentally different mechanism is a more general principle for drug efficacy and selectivity not only for Tyr kinases such as Abl, but also for Ser/Thr kinases. To this end, we chose the Ser/Thr kinase Aurora A and investigated the binding kinetics of three distinct kinase drugs: Danusertib, AT9283, and Gleevec. Aurora A kinase is one of the key regulators of mitotic events, including mitotic entry, centrosome maturation and spindle formation (*Fu et al., 2007*; *Lukasiewicz and Lingle, 2009*; *Marumoto et al., 2005*), as well as assisting in neuronal migration (*Nikonova et al., 2013*). Aurora A has attracted significant attention for the development of targeted agents for cancer because it is overexpressed in a wide range of tumors, including breast, colon, ovary and skin malignancies (*Carvajal et al., 2006*; *Gautschi et al., 2008*; *Katayama and Sen, 2010*; *Lok et al., 2010*; *Marzo and Naval, 2013*). The focus was mainly on ATP-competitive inhibitors, but more recently inhibition by allosteric compounds has also been pursued with the aim of achieving higher selectivity (*Asteriti et al., 2017*; *Bayliss et al., 2017*; *Burgess et al., 2016*; *Janeček et al., 2016*; *McIntyre et al., 2017*). So far, only the clinical significance of Aurora A inhibition by ATP-competitive drugs has been established (*Bavetsias and Linardopoulos, 2015*; *Borisa and Bhatt, 2017*), but little is known about their binding mechanisms. Many high-resolution X-ray structures of Aurora A kinase bound to different inhibitors have been solved (*Bavetsias et al., 2015*; *Dodson et al., 2010*; *Fancelli et al., 2006*; *Ferguson et al., 2017*; *Heron et al., 2006*; *Howard et al., 2009*; *Kilchmann et al., 2016*; *Martin et al., 2012*; *Zhao et al., 2008*), but the selectivity profile of those kinase inhibitors remains very difficult to explain.

The drugs used in this study are small, ATP-competitive inhibitors. Danusertib (PHA739358) and AT9283 were developed for Aurora kinases, whereas Gleevec is selective for the Tyr kinase Abl. Danusertib inhibits all members of the Aurora family with low nanomolar $IC_{50}$ values (13, 79 and 61 nM for Aurora A, B and C, respectively) (*Carpinelli et al., 2007*; *Fraedrich et al., 2012*) and was one of the first Aurora kinase inhibitors to enter phase I and II clinical trials (*Kollareddy et al., 2012*; *Steeghs et al., 2009*). A crystal structure of Danusertib bound to Aurora A kinase shows an inactive kinase with the DFG-loop in the *out* conformation (*Fancelli et al., 2006*). AT9283 inhibits both Aurora A and B with an $IC_{50}$ of 3 nM (*Howard et al., 2009*) and has also entered several clinical trials (*Borisa and Bhatt, 2017*). Interestingly, the crystal structure of Aurora A with AT9283 shows that this drug binds to the DFG-*in*, active conformation of the kinase (*Howard et al., 2009*). Both drugs are high-affinity binders that reportedly bind to a discrete kinase conformation and would allow us to probe for a conformational-selection step. Lastly, we selected Gleevec as a drug that is not selective for Aurora A and should, therefore, have a weaker binding affinity. We reasoned that this choice of inhibitors could reveal general mechanisms underlying drug selectivity and affinity.

The combination of X-ray crystallography, NMR spectroscopy and comprehensive analysis of drug binding and release kinetics delivered a general mechanistic view. Differential drug affinity is not rooted in the overwhelmingly favored paradigm of the DFG-conformation, but instead in the dynamic personality of the kinase that is manifested in conformational changes after drug binding. Notably, such conformational changes have evolved for its natural substrates, and the drugs take advantage of this built-in protein dynamics.

## Results

### Dephosphorylated Aurora A samples both an inactive and active structure

A plethora of X-ray structures and functional assays led to the general notion that dephosphorylated Aurora A and, more universally, Ser/Thr kinases are in an inactive conformation and that phosphorylation or activator binding induces the active structure. A comparison of many X-ray structures of inactive and active forms of Ser/Thr kinases resulted in an elegant proposal of the structural hallmarks for the active state by Taylor and collaborators: the completion of both the regulatory and catalytic spines spanning the N- and C-terminal domains, including the orientation of the DFG-motif (*Kornev and Taylor, 2010*; *2015*). X-ray structures, however, provide merely static snapshots of possible kinase conformations that do not necessarily reflect the situation in solution. In fact, recent experimental data postulate that phosphorylation of Aurora A does not 'lock' the kinase in the active conformation, and that the activation-loop still exhibits conformational dynamics (*Gilburt et al., 2017*; *Ruff et al., 2018*). On the other hand, X-ray crystallography provides high-resolution structural data that cannot readily be obtained from FRET or EPR and IR spectroscopy.

Two crystals from the same crystallization well capture both the inactive and active conformations of dephosphorylated Aurora A bound with AMPPCP (*Figure 1A,B*). As anticipated, the first structure (PDB 4C3R [*Zorba et al., 2014*]) superimposes with the well-known inactive, dephosphorylated Aurora A structure (PDB 1MUO [*Cheetham et al., 2002*]) and the activation loop is not visible as commonly observed for kinases lacking phosphorylation of the activation loop (*Zorba et al., 2014*). The second structure (PDB 6CPF; *Table 1*) adopts the same conformation as the previously published phosphorylated, active structure (PDB 1OL7 [*Bayliss et al., 2003*]) (*Figure 1C*) and the first part of the activation loop could be built, although the B-factors are high. Every hallmark of an active state is seen for this dephosphorylated protein, including the DFG-*in* conformation that is essential for completing the regulatory spine. In contrast, the DFG-loop is in the *out* position for the inactive form of Aurora A (*Figure 1D*, cyan). In the active, non-phosphorylated structure, electron density is seen in the canonical tighter $Mg^{2+}$-binding site, where the metal ion is coordinated to the α- and β-phosphates of AMPPCP and Asp274. The presence of the metal is supported by the CheckMyMetal (*Zheng et al., 2017*) validation, except that the coordination is incomplete. We surmise that two water molecules, not visible in our data, complete the coordination sphere as is seen in several higher-resolution structures. In the inactive structure, no electron density for $Mg^{2+}$ can be identified possibly due to the fact that Asp274 is rotated to the DFG-*out* position and is, therefore, lost as coordination partner. Furthermore, sampling of the active conformation does not depend on

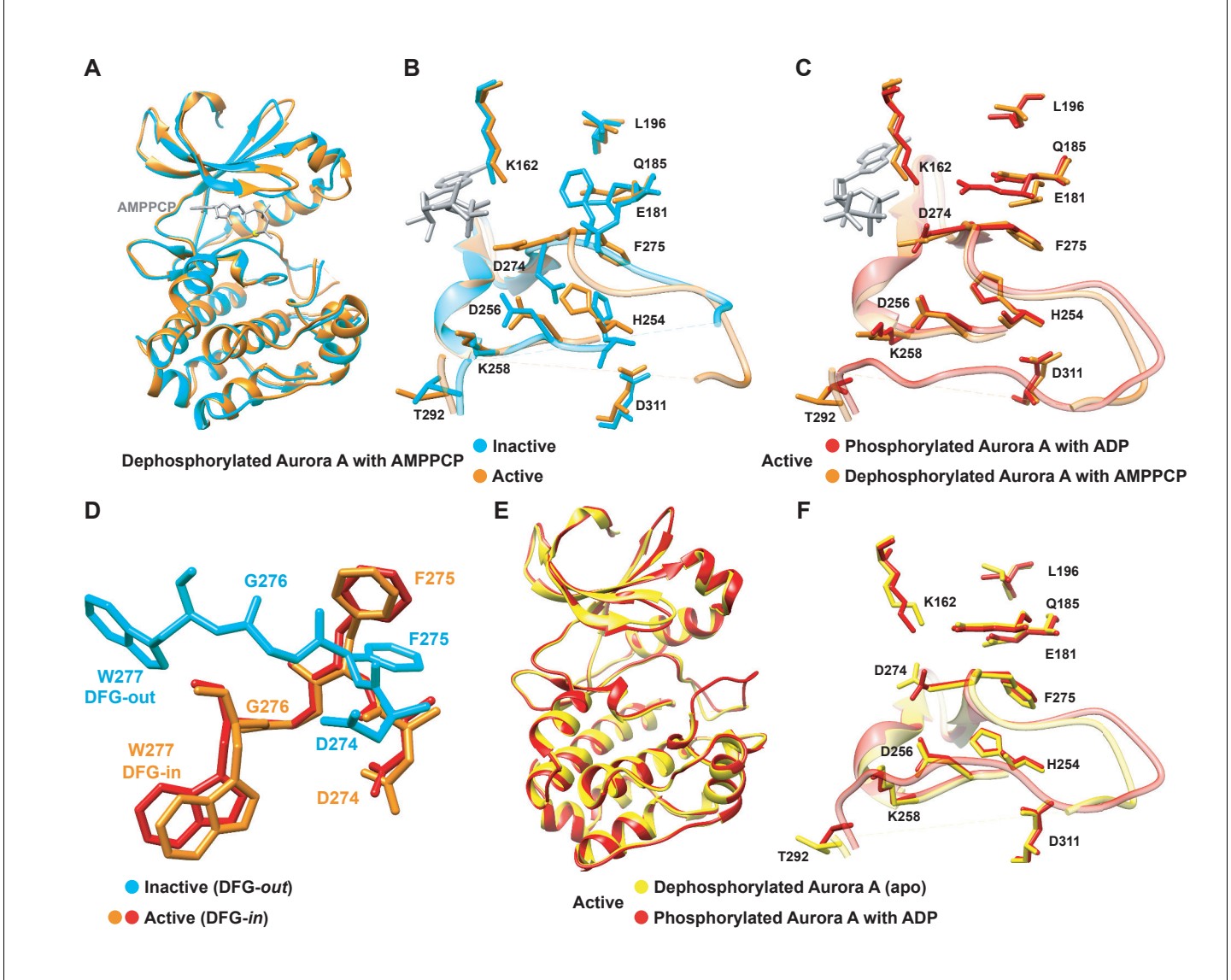

**Figure 1.** Dephosphorylated Aurora A samples both the active and inactive conformation. (A) Superposition of X-ray structures of dephosphorylated Aurora A (residues 122–403) with Mg²⁺·AMPPCP (AMPPCP in gray sticks and magnesium as yellow sphere) in the inactive (cyan, PDB 4C3R [*Zorba et al., 2014*]) and active (orange, PDB 6CPF) state, solved from crystals of the same crystallization well. (B) Zoom-in of (A) to visualize the nucleotide binding region (K162, D274, and E181), the R-spine (L196, Q185, F275, H254, and D311) and the activation loop region (D256, K258, and T292). (C) Same zoom-in as in (B), but dephosphorylated Aurora A in active state (orange) is superimposed with phosphorylated Aurora A (red, PDB 1OL7 [*Bayliss et al., 2003*]). (D) Superposition of the DFG(W) motif in the three states shown in (B) and (C). (E) Superposition of phosphorylated Aurora A in active conformation (red) and apo, dephosphorylated Aurora A also in the active conformation (yellow, PDB 6CPE). (F) Zoom-in of (E) showing the same region as in (B).

DOI: https://doi.org/10.7554/eLife.36656.003

AMPPCP binding as dephosphorylated, apo Aurora A also crystallizes in the active form (PDB 6CPE; *Figure 1E,F* and *Table 1*). Our results are consistent with other crystallographic studies on wild-type, dephosphorylated Aurora A in its apo or nucleotide bound state, where the kinase was also found in the active conformation (*Gustafson et al., 2014*; *Janeček et al., 2016*; *Nowakowski et al., 2002*).

We note that in Aurora kinase sequences a tryptophan residue, Trp277, is immediately following the DFG motif and displays a drastically different orientation whether Aurora A is in an active (DFG-*in*) or inactive (DFG-*out*) conformation (*Figure 1D*). This Trp moiety is unique for the Aurora kinase

**Table 1.** Data collection and refinement statistics for dephosphorylated Aurora A (122-403).

| | apo-Aurora A (6CPE) | Aurora A + AMPPCP (6CPF) | Aurora A + Mb + AT9283 (6CPG) |
|---|---|---|---|
| Data collection | | | |
| Space group | P 61 2 2 | P 61 2 2 | P 21 21 21 |
| Cell dimensions | | | |
| $a, b, c$ (Å) | 80.55, 80.55, 169.79 | 81.75, 81.75, 172.87 | 63.86, 69.7, 175.56 |
| $\alpha, \beta, \gamma$ (°) | 90, 90, 120 | 90, 90, 120 | 90, 90, 90 |
| Resolution (Å) | 84.90–2.45 (2.55–2.45)* | 86.44–2.30 (2.39–2.30)* | 43.14–2.80 (2.87–2.80)* |
| $R_{meas}$ | 0.073 (1.308) | 0.113 (2.260) | 0.189 (1.268) |
| $I/\sigma(I)$ | 15.0 (1.6) | 10.3 (1.3) | 8.9 (1.1) |
| $CC_{1/2}$ | 0.998 (0.711) | 0.997 (0.465) | 0.986 (0.625) |
| Completeness (%) | 99.9 (100) | 100 (100) | 99.2 (98.8) |
| Redundancy | 7.6 (6.3) | 9.7 (7.8) | 5.4 (5.3) |
| | | | |
| Refinement | | | |
| Resolution (Å) | 64.52–2.45 | 54.79–2.30 | 36.17–2.80 |
| No. reflections | 12617 (1224) | 15756 (1527) | 19556 (1845) |
| $R_{work}/R_{free}$ | 0.2151/0.2528 | 0.2179/0.2587 | 0.2792/0.3350 |
| No. atoms | | | |
| Protein | 2035 | 2055 | 5122 |
| Ligand/ion | 11 | 32 | 56 |
| Water | 4 | 6 | |
| $B$ factors | | | |
| Protein | 71.83 | 63.68 | 78.84 |
| Ligand/ion | 75.77 | 76.44 | 81.05 |
| Water | 52.52 | 45.84 | |
| R.m.s. deviations | | | |
| Bond lengths (Å) | 0.005 | 0.004 | 0.003 |
| Bond angles (°) | 0.98 | 0.97 | 0.98 |

The number of crystals for each structure is one for apo-Aurora A and Aurora A + AMPPCP and two crystals for Aurora A + Mb + AT9283.

*Values in parentheses are for highest-resolution shell.

DOI: https://doi.org/10.7554/eLife.36656.004

family in the Ser/Thr kinome and its position is suggested to be important for tuning the substrate specificity (*Chen et al., 2014*). We used this Trp residue as probe to monitor the DFG flip and drug binding in real time as described below.

The fact that the inactive and active states are seen in the crystal implies that both are sampled; however, it does not deliver information about the relative populations or interconversion rates. Therefore, we set out to monitor the conformational exchange of the DFG-*in/out* flip in solution. Owing to the reported importance of the DFG flip for activity, regulation and drug design, there have been extensive efforts to characterize this conformational equilibrium by computation (*Badrinarayan and Sastry, 2014*; *Barakat et al., 2013*; *Meng et al., 2015*; *2017*; *Sarvagalla and Coumar, 2015*; *Shukla et al., 2014*). The general notion of these computational studies is that in the absence of phosphorylation the inactive form of the kinase is most favored, in agreement with experimental evidence. Nevertheless, short-lived excursions to the active state are observed.

As an experimental approach, NMR spectroscopy is an obvious choice; however efforts on several Ser/Thr and Tyr kinases led to the general conclusion that the activation loop, including the DFG motif and most of the active-site residues, cannot be detected due to exchange broadening, and at

best can only be seen after binding of drugs that stabilize conformations (*Campos-Olivas et al., 2011*; *Langer et al., 2004*; *Vajpai et al., 2008*; *Vogtherr et al., 2006*).

[$^1$H-$^{15}$N]-TROSY-HSQC experiments on uniformly $^{15}$N-labeled samples of Aurora A proved to be no exception: many peaks are missing and only three out of four tryptophan side chain indole signals are seen in the 2D spectra of a [$^{15}$N]-Trp labeled sample (*Figure 2A,B*). Therefore, we sought a strategy to overcome this general problem of exchange broadening that hampers the detection of the DFG equilibrium. Aurora A was produced containing 5-fluoro-tryptophan residues to allow for one-dimensional $^{19}$F spectroscopy to deal with exchange broadening while providing sensitivity close to proton NMR (*Kitevski-LeBlanc and Prosser, 2012*). Now, we observe as expected four peaks in our NMR spectra for apo- and AMPPCP-bound wild-type Aurora A (*Figure 2C*). A deconvolution of the

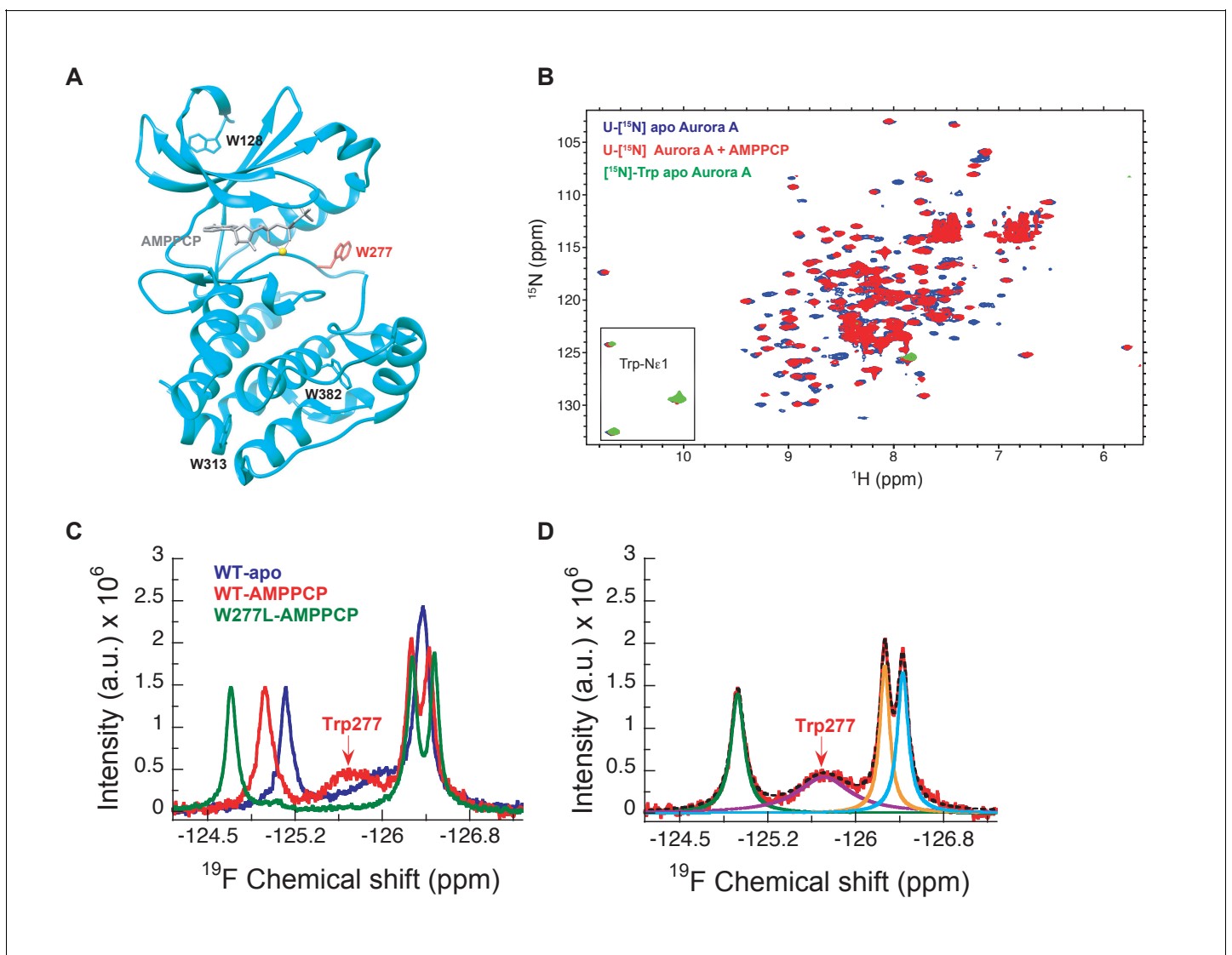

**Figure 2.** NMR spectra indicate extensive dynamics of the DFG-loop. (**A**) The four tryptophan residues in Aurora A are shown on the structure (PDB 4C3R [*Zorba et al., 2014*]) in stick representation; Trp277 in the DFGW-loop is highlighted in red. (**B**) Overlay of [$^1$H-$^{15}$N]-TROSY-HSQC spectra of dephosphorylated Aurora A in its apo-state (U-[$^{15}$N], blue; [$^{15}$N]-Trp, green) and AMPPCP-bound (U-[$^{15}$N], red). Only three instead of the four expected cross peaks for tryptophan side chains are detected. (**C**) $^{19}$F NMR spectra of 5-fluoro-Trp labeled dephosphorylated wild-type Aurora A (apo in blue and AMPPCP-bound in red) and the W277L Aurora A mutant bound to AMPPCP (green). The assignment of Trp277 following the DFG-loop is shown. (**D**) $^{19}$F spectrum of wild-type Aurora A bound to AMPPCP (red) together with its deconvolution into four Lorentzian line shapes, the overall fit is shown as a black, dotted line. The integrals for all four signals are equal, but the linewidth for Trp277 (purple) is approximately 5-fold larger.
DOI: https://doi.org/10.7554/eLife.36656.005

spectrum yields almost identical integral values for all four peaks, whereas the linewidth of one resonance is approximately 5-fold larger (*Figure 2D*, purple signal). This broad peak is a prime candidate to originate from Trp277, directly adjacent to the DFG-loop. The W277L mutation confirmed our hypothesis (*Figure 2C*), and the extensive line broadening of this signal in a one-dimensional spectrum is consistent with its absence in the [$^{1}$H,$^{15}$N]-TROSY-HSQC spectrum. Of note, the W277L mutant is still active, as confirmed by a kinase assay, most likely because this Trp is not conserved in Ser/Thr kinases, where a Leu residue is found at that position for several Ser/Thr family members. Mutating any of the other, more conserved Trp residues resulted in insoluble proteins. The broad line shape for the Trp277 peak hints at severe exchange broadening in the surrounding of the DFG-loop and is consistent with the high B-factors for Trp277 and its neighboring residues observed in all crystal structures described here. Determination of relative populations and rate constants of interconversion is not possible from this data, but this missing piece of information was obtained by stopped-flow kinetics of drug binding.

## Gleevec binding to Aurora A distinguishes conformational selection versus induced-fit mechanisms

Through groundbreaking experiments on the Tyr kinases Abl and Src, the concept of drug selectivity based on the DFG-loop conformation has received considerable attention in kinase drug discovery (*Lovera et al., 2012*; *Treiber and Shah, 2013*). A recent report provides kinetic evidence for such conformational selection, but identifies an induced-fit step after drug binding as the overwhelming contribution for Gleevec selectivity towards Abl compared to Src (*Agafonov et al., 2014*). Here, we ask the obvious question if this mechanism of Gleevec binding to Abl might exemplify a more general mechanism for kinase inhibitors.

To assess which kinetic steps control drug affinity and selectivity, we first studied the binding kinetics for Gleevec to Aurora A by stopped-flow spectroscopy using intrinsic tryptophan fluorescence under degassing conditions to reduce photobleaching. At 25°C, the binding of Gleevec to Aurora A was too fast to be monitored and, therefore, experiments were performed at 10°C. Binding kinetics of Gleevec to Aurora A exhibited biphasic kinetic traces (*Figure 3A*). The first, fast phase is characterized by a decrease in the fluorescence intensity (*Figure 3A,B*), with an observed rate constant, $k_{obs}$, increasing linearly with Gleevec concentration (*Figure 3C*). The slope corresponds to the bimolecular rate constant, $k_2 = 1.1 \pm 0.3$ µM$^{-1}$s$^{-1}$, of Gleevec binding to Aurora A and the dissociation of Gleevec is determined from the intercept, $k_{-2} = 31 \pm 2$ s$^{-1}$ (*Figure 3C*). We note that the parameters for the physical binding step are comparable to the ones obtained for Gleevec binding to Abl (*cf.* $k_2 = 1.5 \pm 0.1$ µM$^{-1}$s$^{-1}$ and $k_{-2} = 25 \pm 6$ s$^{-1}$, measured at 5°C) (*Agafonov et al., 2014*). The second, slow phase exhibits an increase in fluorescence intensity (*Figure 3A*), with the observed rate constant decreasing with Gleevec concentration (*Figure 3D*). The decreasing $k_{obs}$ provides unequivocal evidence of conformational selection, where its rate of interconversion is slower than the rate of ligand dissociation ($k_1 + k_{-1} \ll k_{-2}$). The values of $k_1$ and $k_{-1}$ can be estimated by fitting the data to *Equation 1* and are $0.014 \pm 0.001$ s$^{-1}$ and $0.011 \pm 0.002$ s$^{-1}$, respectively (*Figure 3D*). These rate constants represent the conformational change from DFG-*in* to -*out* and vice versa since Gleevec is a DFG-*out* selective inhibitor due to steric hindrance (*Nagar et al., 2002*; *Schindler et al., 2000*; *Seeliger et al., 2007*).

In order to more rigorously analyze the data and test the model, all time courses of the fluorescence changes were globally fit using the microscopic rate constants determined above as starting values (*Figure 4*) to the model in *Figure 3G*, where also the resulting microscopic rate constants are given. The lack of a conformational transition after drug binding (i.e., induced-fit step) in Aurora A should dramatically decrease drug affinity in comparison to Abl. Indeed, Gleevec binds to Aurora A with a $K_D$ of $24 \pm 7$ µM (*Figure 3F*) compared to the low nM affinity to Abl (*Agafonov et al., 2014*). Two pieces of independent evidence establish that there is indeed no induced-fit step in Gleevec binding to Aurora A: (i) the calculated $K_D$ from the kinetic scheme is in agreement with the macroscopically measured $K_D$ (*cf. Figure 3G and F*), and (ii) the observed $k_{off}$ from the dilution experiment (*Figure 3E*) coincides with the physical dissociation rate (i.e., intercept of the binding plot, $31 \pm 2$ s$^{-1}$, in *Figure 3C*). In summary, the lack of an induced-fit step for Gleevec binding to Aurora A is the major reason for Gleevec's weak binding, and not the DFG-loop conformation or physical drug-binding step, consistent with our earlier results (*Wilson et al., 2015*).

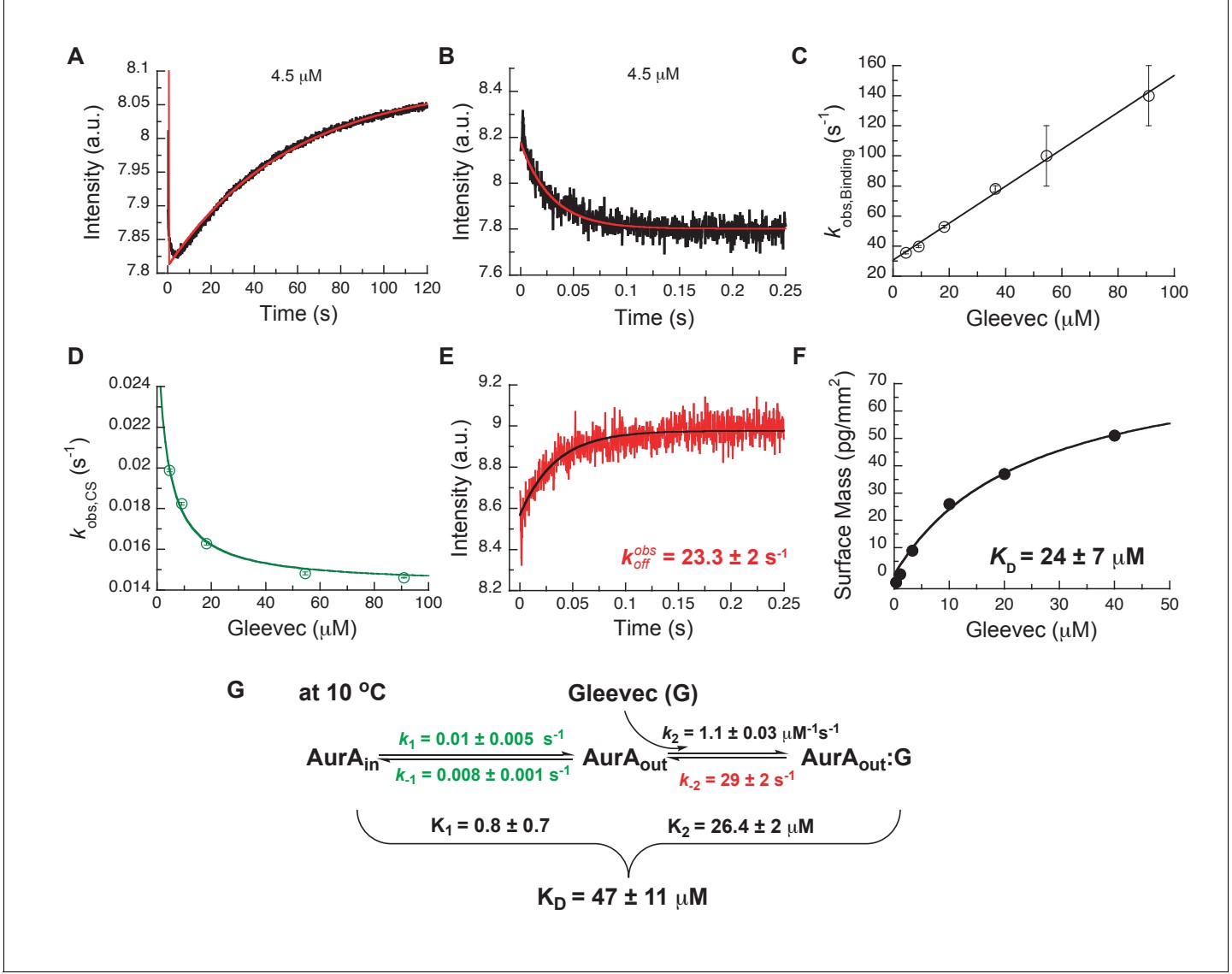

**Figure 3.** Kinetics of Gleevec binding to Aurora A at 10°C measured by stopped-flow Trp fluorescence to dissect all binding steps. (A) Kinetics after mixing 0.5 μM Aurora A with 4.5 μM Gleevec is double exponential with a fast decrease and a slow increase in fluorescence signal. (B) The decrease in fluorescence intensity due to the fast binding phase was completed within 0.25 s. (C) Observed rate constants of fast binding phase were plotted against increasing concentrations of Gleevec ($k_{obs, Binding}$= 1.1 ± 0.3 μM$^{-1}$s$^{-1}$, $k_{diss}$ = 31 ± 2 s$^{-1}$ from the y-intercept). (D) The increase in fluorescence intensity of slow phase (A) is attributed to conformational selection. The plot of $k_{obs, CS}$ of this slow phase *versus* Gleevec concentration was fit to *Equation 1* and yields $k_1$ = 0.014 ± 0.001 s$^{-1}$ and $k_{-1}$ = 0.011 ± 0.002 s$^{-1}$. (E) Dissociation kinetics of pre-incubated solution with 5 μM Aurora A and 5 μM Gleevec measured by stopped-flow fluorescence after an 11-fold dilution of the complex yields the $k_{-2}$ = 23.3 ± 2 s$^{-1}$. (F) The macroscopic dissociation constant ($K_D$) of Gleevec binding to Aurora A measured by Creoptix WAVE. (G) Gleevec (labeled as G) binding scheme to Aurora A corresponds to a two-step binding mechanism: conformational selection followed by the physical binding step. The corresponding microscopic rate constants obtained from the global fit and calculated overall equilibrium and dissociation constants are shown. Fluorescence traces are the average of at least five replicate measurements (n > 5), and error bars and uncertainties given in C-G denote the (propagated) standard deviation in the fitted parameter.

DOI: https://doi.org/10.7554/eLife.36656.006

## Kinetics of Danusertib binding to Aurora A: three-step kinetics with conformational selection and an induced-fit step

Next, we wanted to shed light on why Danusertib, unlike Gleevec, binds very tightly to Aurora A. A high-resolution X-ray structure shows Danusertib bound to Aurora A's active site with its DFG-loop

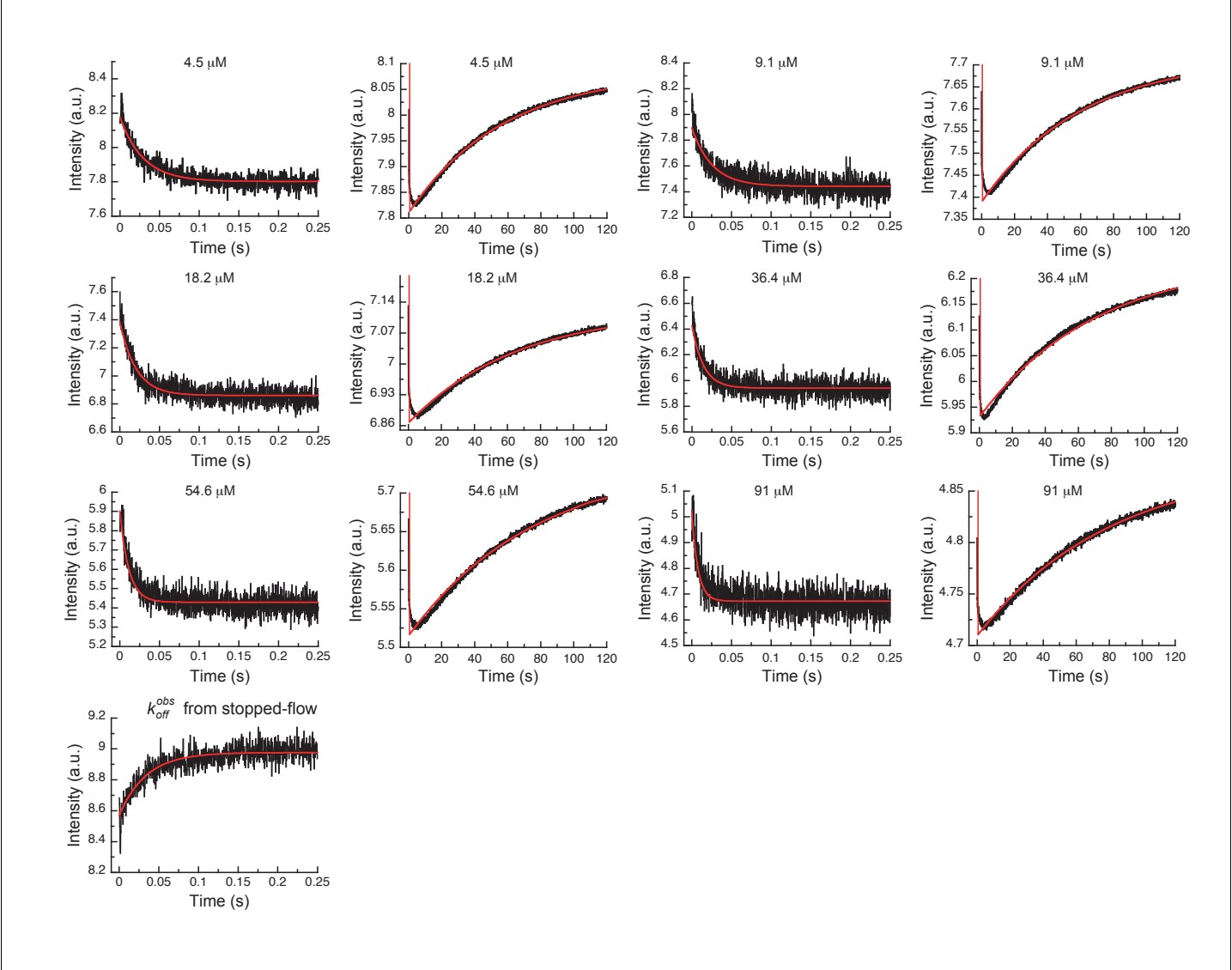

**Figure 4.** Global fits of Gleevec binding- and dissociation-kinetics to Aurora A at 10°C. Fitting of kinetic traces (average, n > 5) of the mixing of 0.5 μM Aurora A with different Gleevec concentrations at two timescales, 0.25 and 120 s, and dissociation kinetics ($k_{off}$) were performed using the KinTek Explorer software with the binding scheme in *Figure 3G*. Red lines show the results of the global fit to the experimental data in black.
DOI: https://doi.org/10.7554/eLife.36656.007

in the *out* conformation (*Figure 5A*) (*Fancelli et al., 2006*), and to rationalize Danusertib's high affinity we measured the kinetics of Danusertib binding to Aurora A directly by stopped-flow experiments at 25°C. An increase in fluorescence intensity was observed at all Danusertib concentrations and showed double-exponential behavior (*Figure 5B*). The dependence of the two observed rates constants on drug concentration is linear for one of them (*Figure 5C*) and non-linear for the other with an apparent plateau at approximately 16 ± 2 s$^{-1}$ (*Figure 5D*). The step with linear inhibitor concentration dependence corresponds to the second-order binding step, whereas a non-linear concentration dependency hints at protein conformational transitions. For a hyperbolic increase of the observed rate with substrate concentrations, one cannot *a priori* differentiate between a conformational selection and an induced fit mechanism. However, conformational selection happens before drug binding, and the intrinsic slow DFG-*in* to DFG-*out* interconversion in Aurora A revealed by Gleevec binding (*Figure 3*) must, therefore, be unaltered. Since the apparent rate of 16 ± 2 s$^{-1}$

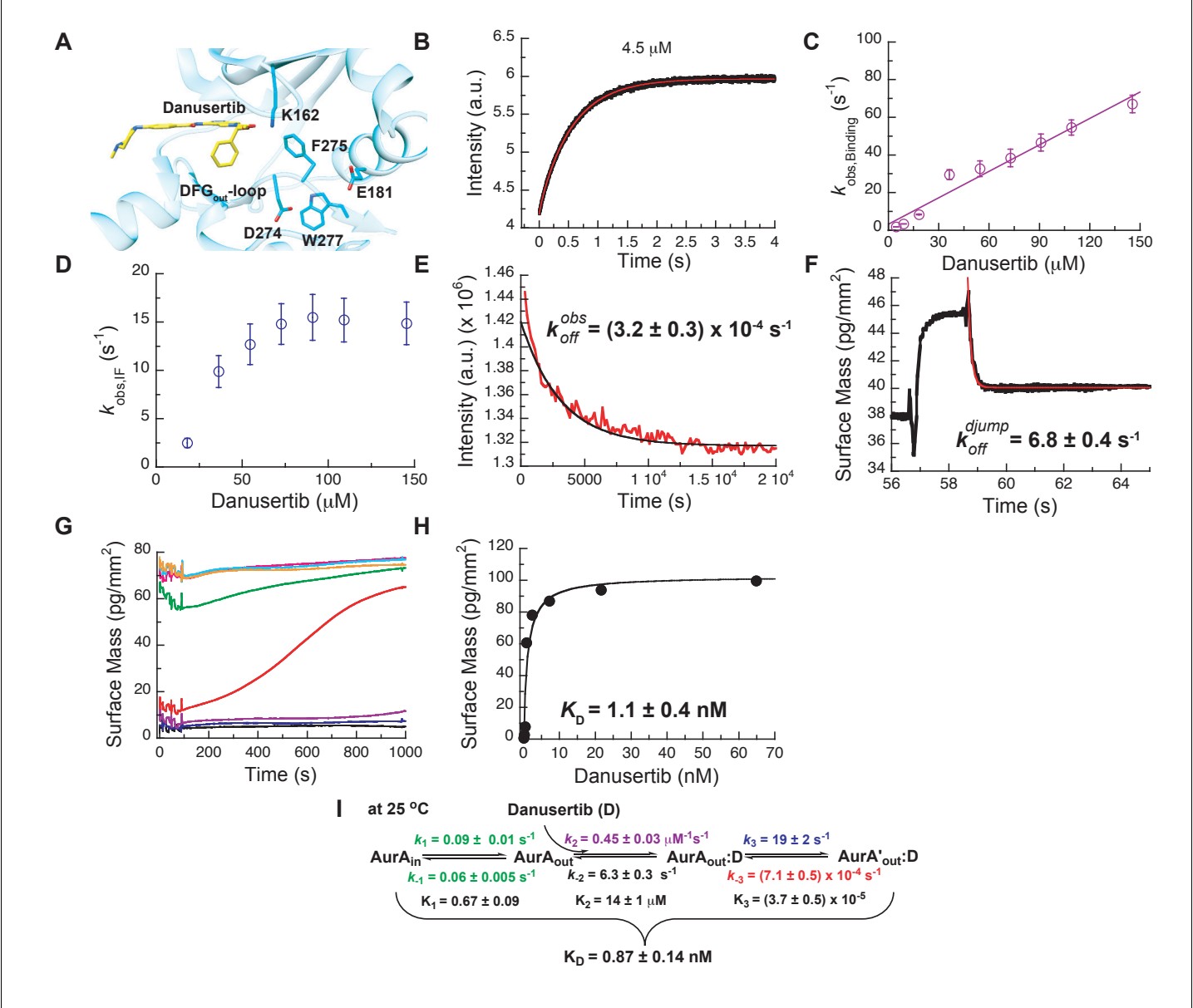

**Figure 5.** Mechanism of Danusertib binding to Aurora A at 25°C. (A) Danusertib bound to the DFG-*out* conformation of Aurora A is shown highlighting important active-site residues in stick representation (PDB 2J50 [*Fancelli et al., 2006*]). (B) The increase in fluorescence upon Danusertib binding is fitted to a double exponential. (C) Plot of $k_{obs,Binding}$ versus the concentration of Danusertib for the fast phase yields $k_2 = 0.4 \pm 0.1$ $\mu M^{-1} s^{-1}$ and $k_{-2} = 4.6 \pm 3$ $s^{-1}$ and the $k_{obs,IF}$ for the slow phase (D) reaches a plateau around $16 \pm 2$ $s^{-1}$. (E) Dissociation of Danusertib from Aurora A at 25°C after a 30-fold dilution of the Aurora A/Danusertib complex measured by Trp-fluorescence quenching and fitting with single exponential gives a value of $k_{-3} = (3.2 \pm 0.3) \times 10^{-4}$ $s^{-1}$. (F) Double-jump experiment (2 s incubation time of 1 $\mu M$ Danusertib to Aurora A followed by 60 s long dissociation step initiated by a wash with buffer) was measured by Creoptix WAVE waveguide interferometry to properly define the value of $k_{-2} = 6.8 \pm 0.4$ $s^{-1}$. (G) Macroscopic dissociation constant ($K_D$) determined by Creoptix WAVE waveguide interferometry: surface-immobilized Aurora A was incubated with various concentrations of Danusertib (0.1 nM (black), 0.2 nM (blue), 0.4 nM (purple), 0.8 nM (red), 2.4 nM (green), 7.2 nM (pink), 21.6 nM (cyan), and 64.8 nM (orange)) and surface mass accumulation was observed until establishment of equilibrium. (H) A plot of the final equilibrium value versus Danusertib concentration yields a $K_D = 1.1 \pm 0.4$ nM. (I) Binding scheme of Danusertib (labeled D) highlighting a three-step binding mechanism, containing both conformational selection and induced-fit step. Red lines in (B, F) and black line in (E) are the results from fitting. Kinetic constants shown in I determined from global fitting (*Figure 6*). Fluorescence traces are the average of at least five replicate measurements (n > 5), and error bars and uncertainties given in C-E, H, and I denote the (propagated) standard deviation in the fitted parameter.

DOI: https://doi.org/10.7554/eLife.36656.008

The following figure supplements are available for figure 5:

**Figure supplement 1.** Additional kinetic experiments to corroborate the three-state binding mechanism for Danusertib to Aurora A.

*Figure 5 continued on next page*

*Figure 5 continued*

DOI: https://doi.org/10.7554/eLife.36656.009

**Figure supplement 2.** Kinetics of Gleevec binding to Aurora A at 25°C to determine DFG-*in*/DFG-*out* equilibrium in apo Aurora A.

DOI: https://doi.org/10.7554/eLife.36656.010

(*Figure 5D*) is two orders of magnitude faster, it can only reflect an induced-fit step (i.e., $k_{obs} = k_3 + k_{-3}$).

So, what happened to the conformational selection step? We hypothesize that the lack of this step in our kinetic traces is due to a too small amplitude of this phase, or not observable because of photobleaching having a bigger effect at the longer measurement times. To lessen potential photobleaching, we reduced the enzyme concentration and increased the temperature to 35°C. Indeed, under these conditions, the slow DFG-*in* to DFG-*out* kinetics were observed as an increase of fluorescence intensity over time with an observed rate constant of approximately 0.1 s$^{-1}$ (*Figure 5—figure supplement 1A*).

While these experiments clearly establish the three-step binding mechanism, it does not provide accurate rate constants for the conformational selection step and it cannot be observed at 25°C where all the other kinetic experiments are performed. To resolve this issue, we repeated the Aurora A–Gleevec experiment at 25°C (*Figure 5—figure supplement 2A,B*) and obtained reliable rate constants ($k_1$ = 0.09 ± 0.01 s$^{-1}$ and $k_{-1}$ = 0.06 ± 0.005 s$^{-1}$) for the conformational selection step in Aurora A, which will be used as 'knowns' in what follows. We hypothesize that the conformational selection step reflects the interconversion between inactive/active conformations and is correlated with the DFG-*out* and -*in* position (*Figure 1*). The following observations support our hypothesis: (i) two crystal structures for the apo-protein show Trp277 in very different environments (*Figure 1E*), (ii) Danusertib has been proposed to selectively bind to the DFG-*out* conformation based on a co-crystal structure (*Figure 5A*) (*Fancelli et al., 2006*), and (iii) the same slow step is observed for binding of both Gleevec and Danusertib.

Next, the dissociation kinetics for Danusertib was measured by fluorescence and appeared to be extremely slow with an observed slow-off rate of (3.2 ± 0.3) × 10$^{-4}$ s$^{-1}$ (*Figure 5E*). Rationalization of complex binding kinetics cannot be done anymore by visual inspection and kinetic intuition, which can, in fact, be misleading. In order to elucidate the correct binding mechanism and obtain accurate kinetic parameters, all kinetic traces were globally fit (*Figure 6*) to the three-step binding scheme (*Figure 5I*). Although global fitting of the binding and dissociation kinetics in KinTek Explorer delivered a value for $k_{-2}$, evaluation of the kinetic scheme with respect to the time traces exposes that $k_{-2}$ is not well determined from our experiments. We therefore designed a double-jump experiment to populate the AurA$_{out}$:D state followed by dissociation to obtain more accurate information on $k_{-2}$. Our stopped-flow machine lacks the capability to perform double mixing and, therefore, the double-jump experiment was performed using a Creoptix WAVE instrument. This label-free methodology uses waveguide interferometry to detect refractive index changes due to alteration in surface mass in a vein similar to Surface Plasmon Resonance (SPR). It is an orthogonal technique that sidesteps notable issues associated with fluorescence methods (e.g., photobleaching and inner-filter effects). In short, after immobilizing Aurora A on a WAVEchip, a high concentration of Danusertib was injected for a short, variable period of time, and dissociation was triggered by flowing buffer through the microfluidics channel to remove the drug. The dissociation kinetics fit to a single exponent with a rate constant, $k_{-2}$, of 6.8 ± 0.4 s$^{-1}$ (*Figure 5F* and *Figure 5—figure supplement 1B*).

We want to discuss a few additional kinetic features. First, the observed rate constant measured in the dilution experiment (*Figure 5E*, $k_{-3}$ = (3.2 ± 0.3) × 10$^{-4}$ s$^{-1}$) is slower than $k_{-3}$ from the global fit ($k_{-3}$ = (7.1 ± 0.5) × 10$^{-4}$ s$^{-1}$), which might seem counterintuitive. The observed rate constant was verified by an additional dilution experiment using Creoptix WAVE ($k_{-3}$ = (2.0 ± 0.6) × 10$^{-4}$ s$^{-1}$, *Figure 5—figure supplement 1C*). The difference in the observed and microscopic rate constant can, however, be fully reconciled by considering the kinetic partitioning for the proposed scheme, as shown in *Figure 6—figure supplement 1*. Second, a powerful and independent validation of the three-step binding mechanism is obtained by comparing the measured overall $K_D$ of Danusertib with the calculated macroscopic $K_D$ from the microscopic rate constants (*Figure 5G,H,I* and *Figure 5—figure supplement 1D*) according to *Equation 4*, which indeed delivers values that are within

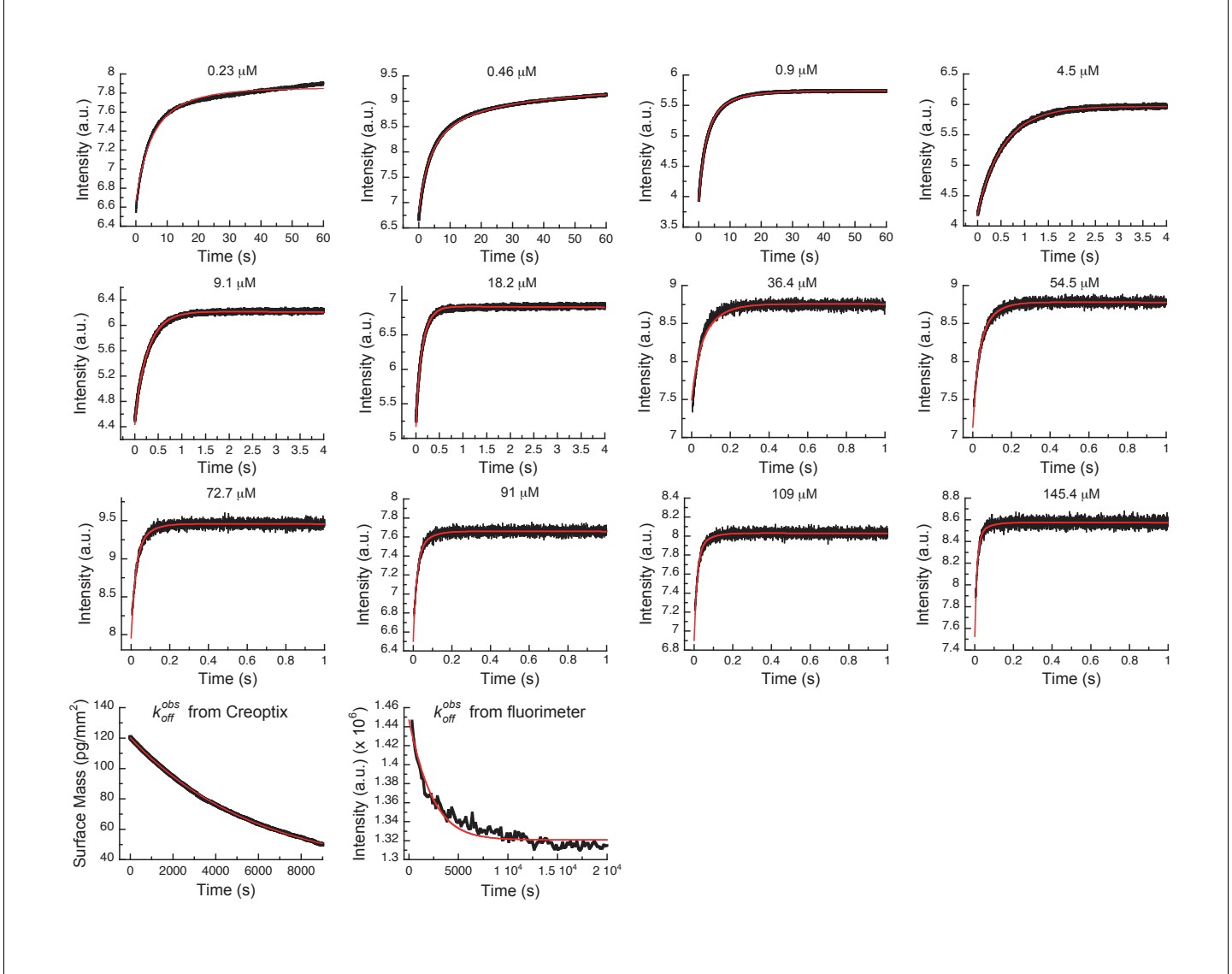

**Figure 6.** Global fits of Danusertib binding and dissociation kinetics to Aurora A at 25°C. Binding kinetics was monitored by stopped-flow fluorescence for different concentrations of Danusertib (indicated) to 0.5 µM Aurora A, and dissociation kinetics ($k_{off}^{obs}$) by Creoptix and fluorimeter (see *Figure 5*). Fluorescence traces are the average of at least five replicate measurements (n > 5). Global fitting was performed using the KinTek Explorer software using the model shown in *Figure 5I*.

DOI: https://doi.org/10.7554/eLife.36656.011

The following figure supplements are available for figure 6:

**Figure supplement 1.** Kinetic partitioning of Aurora A with Danusertib.

DOI: https://doi.org/10.7554/eLife.36656.012

**Figure supplement 2.** Effect of the equilibrium constant for the conformational selection and induced-fit step on the overall $K_D$ for Danusertib.

DOI: https://doi.org/10.7554/eLife.36656.013

experimental error. In addition, our values for $k_2$, $k_{-3}$, and $K_D$ are in good agreement with those reported in a recent study using SPR (*Willemsen-Seegers et al., 2017*).

Our results illuminate trivial but profound principles of binding affinity and lifetime of drug/target complexes: a conformational selection mechanism always weakens the overall inhibitor affinity, while an induced-fit step tightens the affinity depending on how far-shifted the equilibrium in the enzyme/drug complex is (*Equations 2–4*, *Figure 6—figure supplement 2*). For DFG-*out* binders (e.g., Danusertib and Gleevec), the DFG-*in* and -*out* equilibrium weakens the overall affinity 1.6-fold; however,

the conformational change after drug binding results in a four orders of magnitude tighter binding for Danusertib and is the sole reason for its high affinity to Aurora A compared to Gleevec. The dissociation constants for the bimolecular binding step $K_2$ is very similar for both inhibitors. Finally, the lifetime of Danusertib on the target is very long because of the very slow conformational dynamics within the Aurora A/Danusertib complex ($k_{-3} = (7.1 \pm 0.5) \times 10^{-4}$ s$^{-1}$). Earlier examples of protein kinases that also show remarkable slow off-rates, presumably caused by conformational changes, include the epidermal growth factor receptors (*Berezov et al., 2001*; *Wood et al., 2004*) and CDK8 (*Schneider et al., 2013*) amongst others (*Willemsen-Seegers et al., 2017*). To the best of our knowledge, we present here for the first time a detailed stopped-flow kinetics analysis for Aurora A that unequivocally shows the slow off-rate is caused by the conformational change within the drug-bound state, and not the dissociation step.

## Kinetics of AT9283 binding to Aurora A – a surprise

We chose AT9283 as a third inhibitor to characterize the binding mechanism because it has been described as a DFG-*in* binder based on a crystal structure of AT9283 bound to Aurora A (PDB 2W1G, [*Howard et al., 2009*]). We, therefore, anticipated that in its binding kinetics one can now detect the DFG-*out* to DFG-*in* switch. Rapid kinetic experiments of binding AT9283 to Aurora A at 25°C resulted in biphasic traces and both processes showed an increase in fluorescence over time (*Figure 7A*). The $k_{obs}$ for the faster phase ($k_2$) was linearly dependent on drug concentration reflecting the binding step (*Figure 7B*) and $k_{obs}$ for the slower phase ($k_3$) has a limiting value of 0.8 ± 0.2 s$^{-1}$ and is attributed to an induced-fit step (*Figure 7C*). For the conformational selection step (i.e., DFG-*out* to DFG-*in*), a decrease in fluorescence is expected because for the reverse flip observed in the Gleevec and Danusertib experiments, a fluorescence increase was seen (*Figure 3A* and *Figure 5—figure supplement 1A*). However, we could not find any condition (e.g., by varying temperature and ligand concentrations) where such a phase could be observed.

Dissociation is characterized by double-exponential kinetics (*Figure 7D* and *Figure 7—figure supplement 1A*). The fast phase (~38% of the total amplitude change) decays with a rate constant of (1.1 ± 0.02) × 10$^{-2}$ s$^{-1}$, and the slow phase (~62% of the total change in amplitude) has a rate constant of (0.1 ± 0.01) × 10$^{-2}$ s$^{-1}$. To distinguish between the reverse induced-fit step ($k_{-3}$) and the physical dissociation step ($k_{-2}$), a double-jump experiment was performed that unambiguously assigned the faster phase to $k_{-2}$ (*Figure 7E* and *Figure 7—figure supplement 1B*). Our attempts to globally fit all kinetic traces assuming binding to only the DFG-*in* state and using the rate constants for the DFG-loop flip from the Gleevec experiment failed (*Figure 8—figure supplement 1A*). An extended model, where AT9283 can bind to both DFG$_{in/out}$ conformations, followed by a common induced-fit step can also not explain the experimental kinetic traces (*Figure 8—figure supplement 1B*). These failures, together with the lack of a detectable conformational selection step, led to a new model in which both the DFG-*in* and DFG-*out* states can bind AT9283, but only AurA$_{in}$:AT can undergo an induced-fit step (*Figure 7H*). All data can be globally fit to this model (*Figure 8*) and the overall $K_D$ calculated from the corresponding microscopic rate constants (using *Equation 5*) is in good agreement with the experimentally measured $K_D$ (*Figure 7F–H*). Finally, the 10-fold difference between the $k_{-3}$ from the global fit (*Figure 7H*) and the experimentally observed slow off-rate can be reconciled by kinetic partitioning as shown in *Figure 7—figure supplement 1A*.

## Crystal structures of AT9283 bound to Aurora A buttress new binding model

In an effort to structurally verify our model we solved a crystal structure of Aurora A with AT9283 bound and indeed observed the DFG-*out* conformation (PDB 6CPG, *Figure 9B* and *Table 1*), in contrast to the DFG-*in* conformation as previously reported (*Figure 9A*) (*Howard et al., 2009*). Our structure was obtained by co-crystalizing Aurora A with AT9283 and a monobody that binds to the same site as the natural allosteric activator TPX2 (*Figure 9B*). Binding of this monobody shifts Aurora A into an inactive conformation, with the DFG-loop in the *out* conformation. This new structure underscores the plasticity of Aurora A kinase and the ability of AT9283 to bind to a DFG-*out* state, in addition to the previously reported DFG-*in* state.

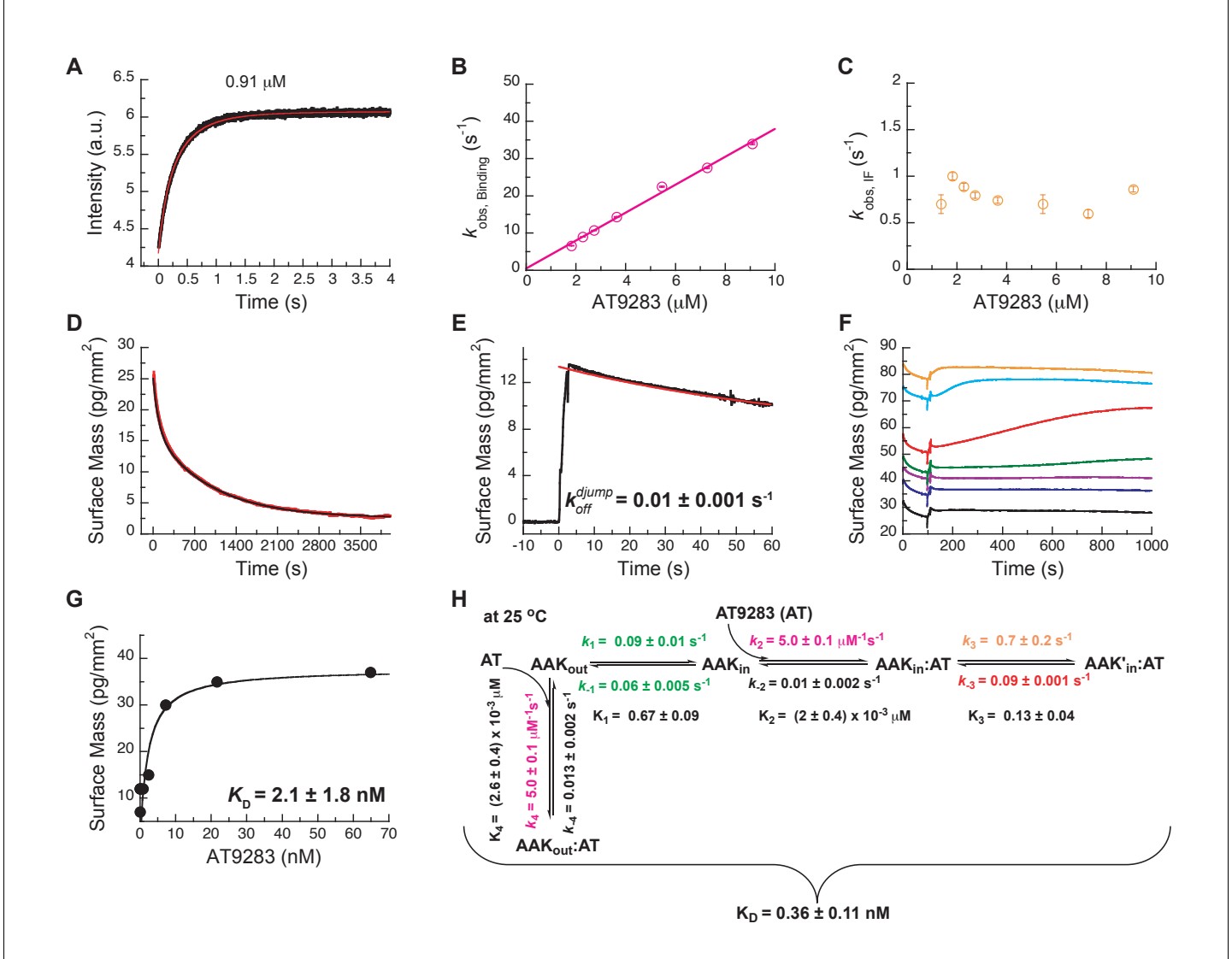

**Figure 7.** Mechanism of AT9283 drug binding to Aurora A at 25°C. (**A**) The increase in fluorescence at 25°C upon AT9283 binding fitted to a double exponential. (**B**) The plot of $k_{obs,Binding}$ versus AT9283 concentration for the fast phase yields $k_2 = 3.4 \pm 0.5 \ \mu M^{-1}s^{-1}$ and an underdetermined intercept ($k_{-2}$) and (**C**) the $k_{obs}$ of the slow phase reaches a plateau around $0.8 \pm 0.2 \ s^{-1}$. (**D**) Dilution of the Aurora A/AT9283 complex formed after 1 hour of incubation. The slow dissociation was measured by Creoptix WAVE waveguide interferometry and fitted with a double exponential with rate constants of $(1.1 \pm 0.02) \times 10^{-2} \ s^{-1}$ and $(0.1 \pm 0.01) \times 10^{-2} \ s^{-1}$. (**E**) Double-jump experiments (1 s incubation time of 1 μM AT9283 to Aurora A followed by 60 s long dissociation step initiated by a wash with buffer) was measured by Creoptix WAVE waveguide interferometry to properly define the value of $k_{-2} = (1.0 \pm 0.1) \times 10^{-2} \ s^{-1}$. (**F**) Macroscopic dissociation constant ($K_D$) determined by Creoptix WAVE waveguide interferometry: surface-immobilized Aurora A was incubated with various concentration of AT9283 (0.03 nM (black), 0.27 nM (blue), 0.8 nM (purple), 2.4 nM (green), 7.2 nM (red), 21.6 nM (cyan), and 64.8 nM (orange)) and surface mass accumulation was observed until establishment of equilibrium. (**G**) A plot of the final equilibrium value versus AT9283 concentration yields a $K_D = 2.1 \pm 1.8 \ nM$. (**H**) Binding scheme for AT9283 (labeled AT) highlighting a four-steps binding mechanism, that contains binding to two different states, a conformational selection mechanism and an induced-fit step. Kinetic constants shown in H were determined from global fitting (see *Figure 8*). Fluorescence traces are the average of at least five replicate measurements (n > 5), and error bars and uncertainties given in B, C, G and H denote the (propagated) standard deviation in the fitted parameter.

DOI: https://doi.org/10.7554/eLife.36656.014

The following figure supplement is available for figure 7:

**Figure supplement 1.** Kinetic partitioning of Aurora A with AT9283.

DOI: https://doi.org/10.7554/eLife.36656.015

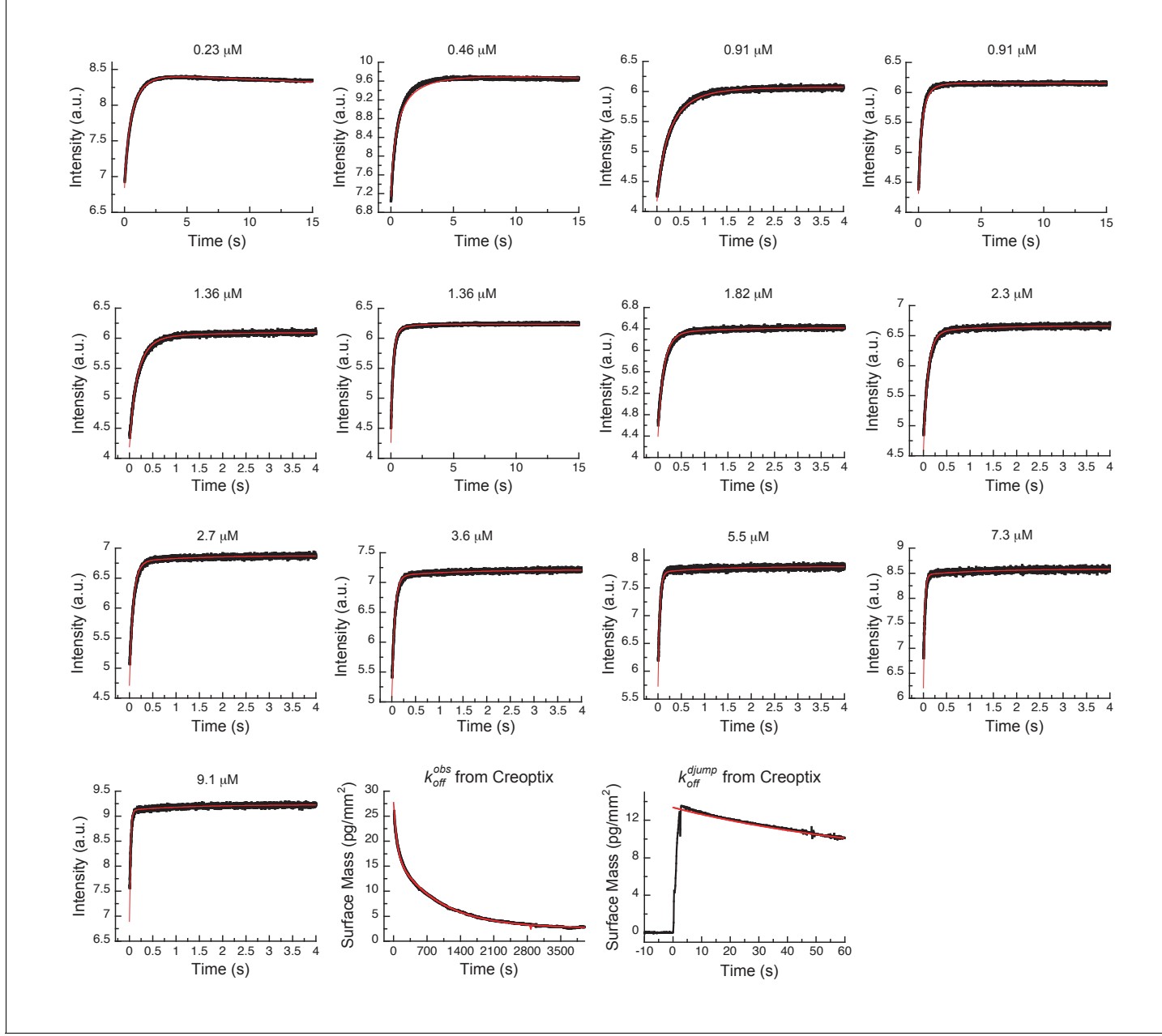

**Figure 8.** Global fits of AT9283 binding and dissociation kinetics to Aurora A at 25°C. Binding kinetics was monitored by stopped-flow fluorescence at different concentrations of AT9283 (indicated) to 0.5 μM Aurora A. Dissociation kinetics were obtained for fully equilibrated drug/kinase complex ($k_{off}^{obs}$) or for the initial encounter complex ($k_{off}^{djump}$) by using a 1 hour or a short 2 s incubation of the kinase with AT9283, respectively, before inducing dissociation by a buffer wash using Creoptix WAVE waveguide interferometry. Global fitting was performed with KinTek Explorer software using the model in *Figure 7H* (reduced $\chi^2$ = 3.2). Fluorescence traces are the average of at least five replicate measurements (n > 5).

DOI: https://doi.org/10.7554/eLife.36656.016

The following figure supplement is available for figure 8:

**Figure supplement 1.** Alternative binding models of AT9283 to Aurora A cannot explain the experimental data.

DOI: https://doi.org/10.7554/eLife.36656.017

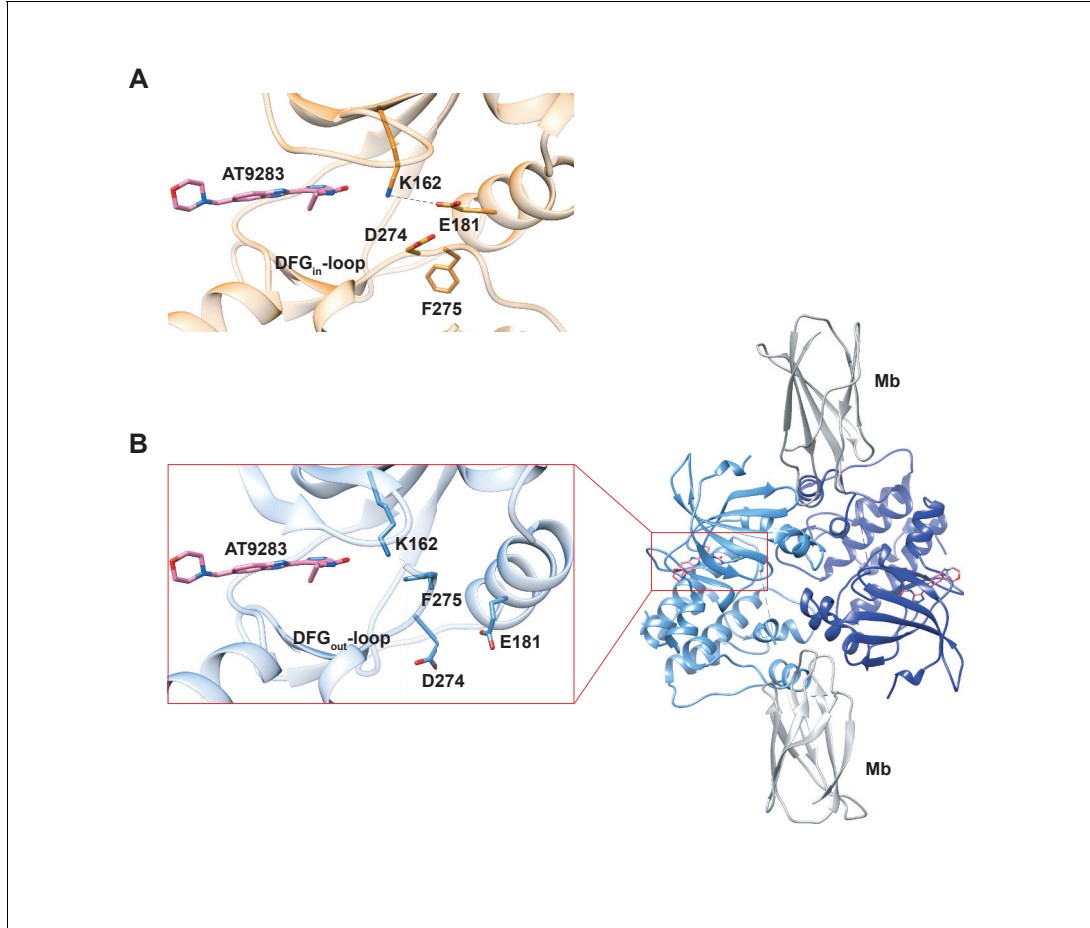

**Figure 9.** X-ray structures of Aurora A bound to inhibitor AT9283 reveal multiple binding modes. (**A**) AT9283 (pink) bound to the active site of Aurora A (PDB 2W1G, [*Howard et al., 2009*]) shows the DFG$_{in}$-loop conformation and a salt bridge between K162 and E181. (**B**) Aurora A dimer (light and dark blue ribbon) in complex with AT9283 (pink) and inhibiting monobody (Mb, grey), showing DFG$_{out}$-loop and broken K162 and E181 salt bridge (PDB 6CPG).

DOI: https://doi.org/10.7554/eLife.36656.018

Thus, our structural and kinetic data together support that AT9283 can bind to both DFG-*in* and DFG-*out* state of Aurora A, and emphasizes the need for caution when interpreting single X-ray structures.

## Inhibitors take advantage of built-in dynamics for ATP binding

We finally compared the binding kinetics of the ATP-competitive inhibitors described above with the natural kinase substrate, ATP (*Figure 10*). In order to measure stopped-flow kinetics for ATP binding, FRET was measured by exciting Trp residues in Aurora A and detecting fluorescence transfer to the ATP-analogue mant-ATP (*Lemaire et al., 2006*; *Ni et al., 2000*). The binding of mant-ATP to Aurora A showed biphasic kinetic traces (*Figure 10A*) that describe the physical binding step (i.e., linear dependence on mant-ATP concentration; *Figure 10B*) and the induced-fit step (*Figure 10C*). The observed rate constant approaches a maximum value defined by the sum of $k_3 + k_{-3}$ (*Figure 10C*) and the intercept can be estimated to be $k_{-3}$ and is consistent with the value obtained from the $k_{off}$ experiment (*Figure 10D*). We find that mant-ATP can bind to both the DFG-*in* and -*out* conformations, consistent with our nucleotide-bound crystal structures (*Figure 1A–D*) and recent single-molecule fluorescence spectroscopy data that indicates that nucleotide binding does not significantly affect this equilibrium (*Cyphers et al., 2017*). To confirm the model, the kinetic data were globally fit to a two-step binding mechanism (*Figure 10G,H*). The calculated $K_D$ from the

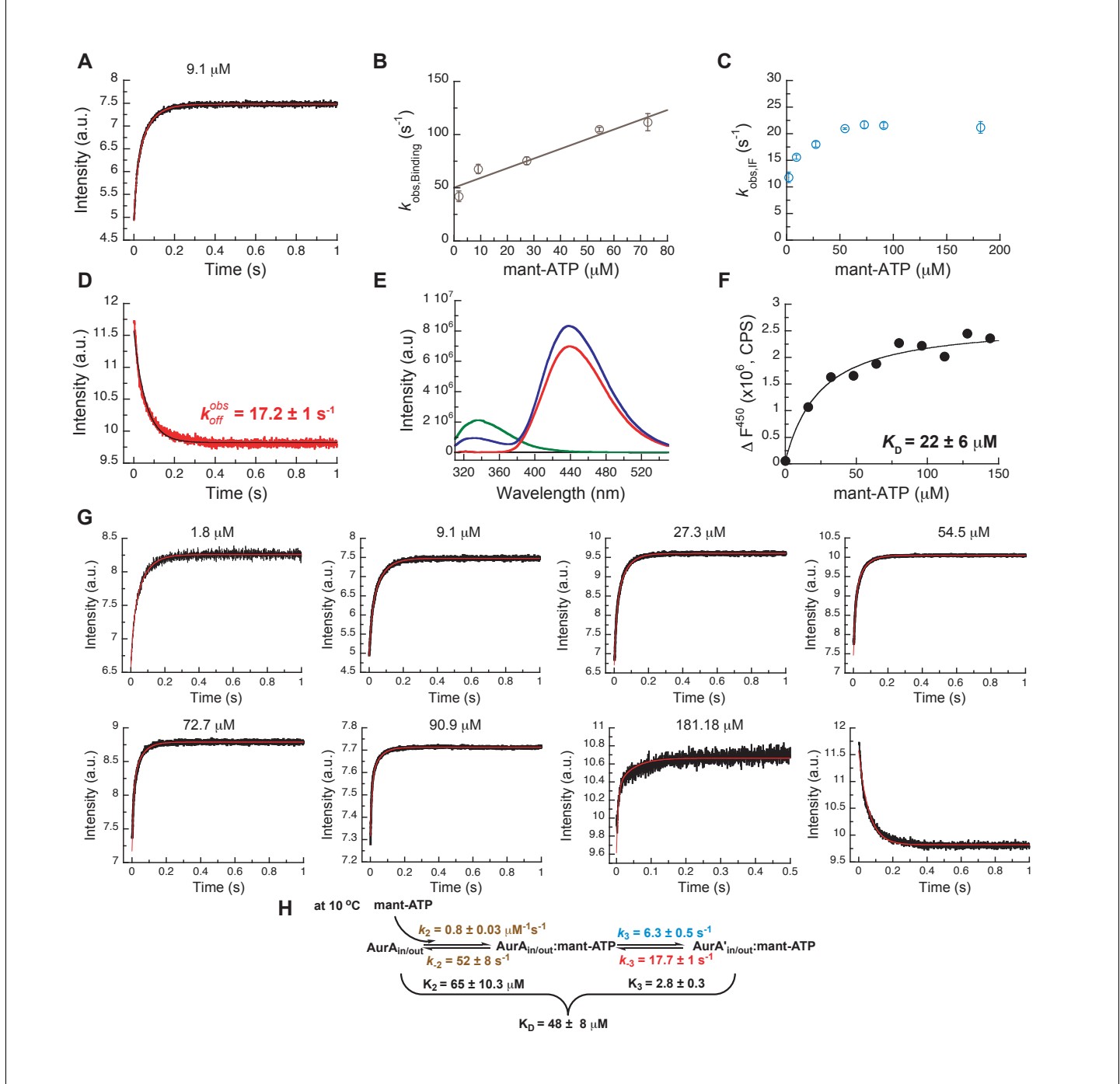

**Figure 10.** Mechanism of ATP binding to Aurora A at 10°C. (A) Binding of mant-ATP to Aurora A was followed by an increase in fluorescence with biphasic kinetics. The plot of $k_{obs}$ versus concentration of mant-ATP of the fast phase (B) yields $k_2 = 0.8 \pm 0.2\ \mu M^{-1}s^{-1}$ and $k_{-2} = 50 \pm 8\ s^{-1}$ and the slow phase (C) reached a plateau around $21 \pm 1\ s^{-1}$ ($k_3 + k_{-3}$). (D) Dissociation kinetics of 10 μM Aurora A/10 μM mant-ATP complex was measured after a 11-fold dilution into buffer and yields $k_{off}^{obs} = 17.2 \pm 1\ s^{-1}$. (E, F) Macroscopic dissociation constant of Aurora A with mant-ATP measured by fluorescence energy transfer. (E) Emission spectra (excitation at 290 nm) of 1 μM Aurora A (green), 160 μM mant-ATP (red), and 1 μM Aurora A/160 μM mant-ATP (blue). (F) The change in fluorescence at 450 nm ($\Delta F^{450}$) versus mant-ATP concentrations yields $K_D = 22 \pm 6\ \mu M$. (G) Global fitting (red) of all kinetics data (black) in KinTek Explorer to the binding scheme shown in (H) results in the kinetic constants given in the scheme and an overall $K_D = 48 \pm 8\ \mu M$, calculated from all rate constants. Fluorescence traces are the average of at least five replicate measurements (n > 5), and error bars and uncertainties given in B, C, D, F, and H denote the (propagated) standard deviation in the fitted parameter.

DOI: https://doi.org/10.7554/eLife.36656.019

corresponding microscopic rate constants (*Figure 10H*) is comparable with the experimental macroscopic $K_D$ obtained from a titration experiment (*Figure 10E,F*).

The presence of an induced-fit step for the natural substrate ATP suggests that such conformational change after ligand binding is a built-in property of the enzyme. In other words, inhibitors take advantage of the inherent plasticity of the enzyme that is required for its activity and regulation. The main difference between ATP and inhibitor binding is the rate constant for the reverse induced-fit step ($k_{-3}$). In the case of ATP, this rate is much faster and, therefore, does not significantly increase the overall affinity. Faster conformational changes and weaker binding are of course prerequisites for efficient turnover; whereas slow conformational changes, particularly the reverse induced-fit step, are at the heart of action for an efficient drug, because it results in tight binding and a long lifetime on the target. In summary, binding of different ligands to the ATP-binding site, such as nucleotides or ATP-competitive inhibitors, is comprised of the physical binding step followed by an induced-fit step. By definition, it is the nature of the induced-fit step that varies for the different ligands since it happens as a result of ligand binding.

## Discussion

Characterizing the detailed kinetic mechanisms of drug binding is not just an academic exercise but delivers fundamental knowledge for developing selective inhibitors with high affinity. An induced-fit step turns out to be key for all tight-binding inhibitors studied. From our results on Aurora A kinase presented here and earlier data on Tyrosine-kinases (*Agafonov et al., 2014*; *Wilson et al., 2015*), we propose that this may be a general mechanism for different kinases and multiple inhibitors, thereby providing a platform for future computational and experimental efforts in rational drug design. Albeit, we note that verification of this proposition requires a larger sampling of small molecules and different protein kinases throughout the kinome.

The 'use' of a highly-skewed equilibrium towards E*:D for a promising drug is logical for the following reasons: (i) it increases the affinity for the drug by this coupled equilibrium, (ii) it prolongs the residence time of the drug on the target due to the often slow reverse rate, (iii) it is specific for each drug as it happens after the drug binding, and (iv) it can add selectivity for the targets because it likely involves residues more remote from the active site. An increased drug residence time has significant pharmacological advantages as it can lead to a prolonged biological effect, a decrease of side effects, and a lower risk of metabolic drug modification. Such inhibitors have long been described as slow tight-binding inhibitors (*Copeland, 2016*; *Copeland et al., 2006*). The concept of the advantageous roles of induced-fit steps is based on simple thermodynamics and protein flexibility, and is, therefore, likely of relevance for drug design to other targets outside of the kinome.

Additionally, our data provides unique insight into the extensively discussed DFG flip. Combining X-ray crystallography, NMR spectroscopy and stopped-flow kinetics of drug binding establish the nature of this DFG flip both structurally, thermodynamically and kinetically, and resolves the long-standing question of its role for drug affinity and selectivity. Selective binding of a specific DFG-state by Gleevec has been first proposed as the reason for selectivity towards Abl. This conformational selection principle has ever since been at the center of drug discovery for many kinases, including Aurora A (*Badrinarayan and Sastry, 2014*; *Liu and Gray, 2006*). Based on our results, we argue that conformational selection of the DFG-state by ATP-competitive inhibitors is a mistakenly pursued concept in drug design for the following reasons: (i) conformational selection by definition weakens the overall ligand affinity, (ii) active site binders are automatically inhibitors, therefore selective binding to a specific DFG-state has no advantage (*Badrinarayan and Sastry, 2014*; *Liu and Gray, 2006*), (iii) kinases interconvert between both states. High selectivity gained by DFG-state selective binding could only be achieved in the scenario of a highly skewed population towards the binding-competent state for one kinase relative to all others, which is unfounded.

Our results exemplify why rational drug design is so challenging. The characterization of the complete free-energy landscape of drug binding is needed, which will require more sophisticated computational approaches guided by experimental data such as provided in our study. A good illustration of this point are the computational reports that focused on the DFG flip as a key determinant drug selectivity (*Badrinarayan and Sastry, 2014*) that now have been ruled out by our kinetic measurements. Our data suggest that future design efforts should be focusing on understanding and exploiting induced-fit steps. To this end, the different dynamic personalities of kinases or, more

general, drug targets need to be investigated at atomic resolution and used to guide small-molecule design. The findings presented here are encouraging for developing selective inhibitors even for kinases with very similar folds and drug binding pockets since the action does not happen on a single structural element of the protein, but on a complex energy landscape that is unique to each kinase.

## Materials and methods

### Cloning, expression and purification of dephosphorylated Aurora A (122-403) and inhibiting monobody

Dephosphorylated Aurora A proteins were expressed and purified as described before (*Zorba et al., 2014*) and analyzed by mass spectrometry to confirm their phosphorylation state. The W227L mutant was generated using the QuickChange Lightning site-directed mutagenesis kit (Agilent).

U-[$^{15}$N] Aurora A was obtained by growing *E. coli* BL21(DE3) (New England Biolabs) in M9 minimal medium containing 1 g/L $^{15}$NH$_4$Cl (Cambridge Isotope Laboratories, Tewksbury, MA, USA) and 5 g/L D-glucose as the sole nitrogen and carbon source, respectively. [$^{15}$N]-Trp labeled wild-type Aurora A was obtained using the standard M9 minimal medium, complemented with all amino acids (0.5 g/L) with the exception of tryptophan. One hour prior to induction, 30 mg/L of $^{15}$N2-L-Trp (NLM-800; Cambridge Isotope Laboratories, Tewksbury, MA, USA) was added to the medium. Similarly, to obtain samples of wild-type and W277L Aurora A containing 5-fluoro-tryptophan, bacterial growth was performed in unlabeled M9 medium containing all amino acids (0.5 g/L) except for tryptophan. One hour before protein induction, the medium was supplemented with 30 mg/L of 5-fluoro-DL-tryptophan (Sigma-Aldrich) (*Crowley et al., 2012*). NMR samples contained 200–300 µM Aurora A in 50 mM HEPES, pH 7.3, 50 mM NaCl, 20 mM MgCl$_2$, 5 mM TCEP, 2 M TMAO and 10% (v/v) D$_2$O.

Inhibiting monobody used for co-crystallization with Aurora A and AT9283 was expressed in *E. coli* BL21(DE3) cells harboring the plasmid pHBT containing His$_6$-tagged-Mb. A culture of TB media containing 50 µg/mL kanamycin that was grown overnight at 37°C was added to 1 L of TB media with 50 µg/mL kanamycin to get a starting OD$_{600}$ of ~0.2. This culture was grown at 37°C until the OD$_{600}$ reached ~0.8. Protein expression was induced by 0.6 mM IPTG at 18°C for 13–15 h and cells were harvested by centrifugation. The cell pellet was resuspended in binding buffer (50 mM Tris-HCl, pH 8.0, 300 mM NaCl, 20 mM imidazole, 20 mM MgCl$_2$, 10% glycerol) containing 0.5 mg/mL lysozyme, 5 µg/mL DNase, and 1x EDTA-free protease inhibitor cocktail. Cells were ruptured by sonication on ice then centrifuged at 18,000 rpm at 4°C for 1 h. The supernatant was loaded onto His-Trap$^{TM}$ HP (GE Healthcare) after filtration using 0.2 µm filtering unit. The pellet was resuspended with GuHCl buffer (20 mM Tris-HCl, pH 8.0, 6 M GuHCl) and allowed to rotate on wheel for 10 min at 4°C and spun down again. The supernatant was passed through 0.2 µm filtering unit and loaded onto HisTrap HP column previously loaded with soluble fraction and pre-equilibrated with GuHCl buffer. Refolding monobody on-column was achieved by washing the HisTrap HP column with five column volumes (CV) of GuHCl buffer, followed by 5 CV of Triton-X buffer (binding buffer + 0.1% Triton X-100), then 5 CV of β-cyclodextrin buffer (binding buffer + 5 mM β-cyclodextrin), and finally 5 CV of binding buffer. Monobody was eluted with 100% of elution buffer (binding buffer + 500 mM imidazole). The protein was dialyzed overnight in gel-filtration buffer (20 mM Tris-HCl, pH 7.5, 200 mM NaCl, 20 mM MgCl$_2$, 5 mM TCEP, 10% glycerol) in the presence of TEV protease (1:40 TEVP: Mb molar ratio). After dialysis, the TEV-cleaved monobody was passed through HisTrap HP column again. The flow-through containing TEV-cleaved monobody was collected and concentrated before loading onto Superdex 200 26/60 gel-filtration column pre-equilibrated with the gel-filtration buffer. The monobody was flash-frozen and stored in −80°C until use.

### X-ray crystallography

Crystals of dephosphorylated (deP) Aurora A$^{122-403}$ + AMPPCP were obtained by mixing 570 µM (18 mg/mL) deP Aurora A$^{122-403}$ and 1 mM AMPPCP in a 2:1 ratio with mother liquor (0.2 M ammonium sulfate, 0.2 M Tris-HCl, pH 7.50, 30% (w/v) PEG-3350). The crystals were grown at 18°C by vapor diffusion using the hanging-drop method. The protein used for the crystallization was in storage buffer (20 mM Tris-HCl, pH 7.5, 200 mM NaCl, 10% (v/v) glycerol, 20 mM MgCl$_2$, 1 mM TCEP);

AMPPCP was freshly prepared before use in the same buffer. Crystals were flash-frozen in liquid nitrogen prior to shipping. Crystals of apo, deP Aurora A$^{122-403}$ were grown at 18°C by vapor diffusion using the sitting-drop method (96-well plate). A 1:1 ratio of protein to mother liquor was obtained by combining 0.5 μL of 300 μM (10 mg/mL) deP Aurora A$^{122-403}$ in 50 mM HEPES, pH 7.3, 500 mM ammonium acetate, 1 mM MgCl$_2$, 5 mM TCEP) with 0.5 μL of 0.15 M Tris-HCl, pH 7.5, 0.15 M ammonium sulfate, 35% (w/v) PEG-3350. Crystals were soaked for 10–20 s in cryo buffer (20% (w/v) PEG-400, 20% ethylene glycol, 10% water and 50% mother liquor) before flash-freezing in liquid nitrogen. The complex between Aurora A$^{122-403}$, inhibiting monobody (Mb) and AT9283 was crystallized at 18°C by vapor diffusion using the sitting-drop method. In short, a 1:1 ratio of protein mixture to mother liquor was obtained by combining 0.5 μL of sample [240 μM deP Aurora A$^{122-403}$ + 1.0 mM AT9283 + 250 μM Mb] with 0.5 μL of mother liquor [0.1 M Bis-Tris, pH 5.5, 0.2 M magnesium chloride, 19% (w/v) PEG-3350]. Crystals were soaked for 10–20 s in cryo buffer (17.5% (w/v) PEG-400, 17.5% ethylene glycol, 45% water and 20% mother liquor) before flash-freezing in liquid nitrogen.

Diffraction data were collected at 100 K at the Advanced Light Source (Lawrence Berkeley National Laboratory) beamlines ALS 8.2.1 (apo-AurA and AurA + Mb + AT9283) and 8.2.2 (AurA + AMPPCP) with a collection wavelength of 1.00 Å.

Data were indexed and integrated using iMOSFLM (*Battye et al., 2011*) for apo/AMPPCP-bound Aurora A and Xia2 (*Winter, 2010*) using XDS (*Kabsch, 2010*) for the Aurora A/Mb/AT9283 complex, respectively. Data were scaled and merged with AIMLESS (*Evans and Murshudov, 2013*), in the case of Aurora A/Mb/AT9283 two data separate data sets were merged. All software was used within the CCP4 software suite (*Winn et al., 2011*).

As initial search models 1MQ4 (*Nowakowski et al., 2002*) and 3K2M (*Wojcik et al., 2010*) were used for Aurora A and monobody, respectively, and molecular replacement was performed using Phaser (*McCoy et al., 2007*). The molecules were placed in the unit cell using the ACHESYM webserver (*Kowiel et al., 2014*). Iterative refinements were carried out with PHENIX (*Adams et al., 2010*), using rosetta.refine (*DiMaio et al., 2013*) and phenix.refine (*Afonine et al., 2012*), and manual rebuilding was performed in Coot (*Emsley and Cowtan, 2004*; *Emsley et al., 2010*).

Structure validation was performed using MolProbity (*Chen et al., 2010*) and yielded the statistics given below. The Ramachandran statistics for dephosphorylated apo (AMPPCP-bound) Aurora A are: favored: 93.65 (94.90)%, allowed 5.95 (4.71)%, outliers: 0.4 (0.39)%; 0.48 (0.0)% rotamer outliers and an all-atom clashscore of 4.45 (2.44). For the Aurora A/Mb/AT9283 complex, the Ramachandran statistics are: favored: 92.64%, allowed 7.06%, outliers: 0.3%; 0.0% rotamer outliers and an all-atom clashscore of 2.81. We note that the B-factors for the monobodies in the complex of Aurora A/Mb/AT9283 are rather high, indicating significantly flexibility in the parts that are not part of the binding interface with Aurora A.

The data collection and refinement statistics are given in *Table 1*. Structure factors and refined models have been deposited in the PDB under accession codes: 6CPE (apo Aurora A), 6CPF (Aurora A + AMPPCP) and 6CPG (Aurora A/Mb/AT9283).

All figures were generated using Chimera (*Pettersen et al., 2004*).

## NMR spectroscopy

All $^{19}$F NMR experiments were performed at 35°C on a Varian Unity Inova 500 MHz spectrometer, equipped with a $^1$H/$^{19}$F switchable probe tuned to fluorine (90° pulse width of 12 μs). All 1D $^{19}$F spectra were recorded with a spectral width of ~60 ppm and a maximum evolution time of 0.25 s. An interscan delay of 1.5 s was used with 5000 scans per transients, giving rise to a total acquisition time of 2.5 h per spectrum. To remove background signal from the probe and avoid baseline distortions, data acquisition was started after a ~100 μs delay (using the 'delacq' macro) and appropriate shifting of the data followed by backward linear prediction was performed. The data were apodized with an exponential filter (2.5 Hz line broadening) and zero-filled before Fourier transform. To improve the signal-to-noise ratio several data sets were recorded consecutively and, provided that the sample remained stable, added together after processing (two for apo Aurora A, four for Aurora A + AMPPCP, and five for W277L + AMPPCP, respectively). $^{19}$F chemical shifts were referenced externally to trifluoroacetic acid (TFA) at −76.55 ppm.

[$^1$H-$^{15}$N]-TROSY-HSQC experiments were recorded at 25°C on an Agilent DD2 600 MHz four-channel spectrometer equipped with a triple-resonance cryogenically cooled probe-head. Typically,

115–128 ($^{15}$N) × 512 ($^{1}$H) complex points, with maximum evolution times equal to 48.5–64 ($^{15}$N) × 64 ($^{1}$H) ms. An interscan delay of 1.0 s was used along with 32 or 56 scans per transient, giving rise to a net acquisition time 1.5–2.5 h for each experiment. To improve the signal-to-noise ratio several data sets were recorded consecutively and, provided that the sample remained stable, added together after processing (typically three data sets per sample).

All data sets were processed with the NMRPipe/NMRDraw software package (*Delaglio et al., 1995*) and 2D spectra were visualized using Sparky (*Goddard, 2008*). Deconvolution of the $^{19}$F spectra and line shape fitting was performed using the Python package nmrglue (*Helmus and Jaroniec, 2013*).

## Kinetics experiments of Aurora A with Gleevec, Danusertib, and AT9283

### Stopped-flow experiments

Intrinsic tryptophan fluorescence spectroscopy was used to monitor drug binding kinetics to Aurora A. All experiments were performed at 25°C, except for the Gleevec kinetics that were measured at 10°C (unless otherwise stated) because the binding of Gleevec to Aurora A is too fast, $k_{obs,Binding}$. Stock solutions of 200 mM Danusertib, 200 mM AT9283 and 50 mM Gleevec (all purchased from Selleck Chemicals, http://www.selleckchem.com) were prepared in 100% DMSO were and stored at −80°C until used. Aurora A used in the kinetic experiments was dephosphorylated Aurora A as determined by mass spectrometry, Western blot and activity experiments (data not shown). The rapid kinetics were studied using a stopped-flow spectrophotometer (SX20 series from Applied Photophysics Ltd). The flow system was made anaerobic by rinsing with degassed buffer comprised of 50 mM HEPES, 50 mM NaCl, 20 mM MgCl$_2$, 5 mM TCEP, 5% DMSO, pH 7.30 to minimize photobleaching. The stock solutions of Aurora A and all drugs were made anaerobic by degassing with ThermoVac (MicroCal) at the desired temperature. In general, a solution of 5 µM Aurora A was loaded in one syringe and quickly mixed with drug, prepared in the same buffer, in the other syringe (mixing ratio 1:10). A significant increase or decrease in the fluorescence intensity of Aurora A (excitation at 295 nm, emission cut-off at 320 nm) can be observed due to the drug binding. For each drug concentration, at least five replicate measurements were made and these transients were averaged. Analysis was performed by fitting the individual trace to exponential equations using Pro-Data Viewer (Applied Photophysics Ltd) or with Kinesyst 3 software (TgK Scientific) and error bars denote the standard errors as obtained from the fit. KaleidaGraph version 4.5.3 (Synergy) was used for data analysis and plotting. All kinetic data were globally fitted in KinTek Explorer software (*Johnson, 2009b, 2009a*).

Under the rapid equilibrium approximation, the binding and dissociation steps of Gleevec to Aurora A are fast compared to conformational selection, therefore the value of $k_1$ and $k_{-1}$ can be estimated according to *Equation 1*:

$$k_{obs} = \frac{k_{-1}}{1 + \left( \frac{[\text{Gleevec}]}{[\text{Aurora A}] + \left( \frac{k_{-2}}{k_2} \right)} \right)} + k_1 \quad (1)$$

where $k_1$ and $k_{-1}$ represent the conformational change from DFG-*in* to -*out* and vice versa, respectively. The approximate values of $k_1$ and $k_{-1}$ obtained from fitting to this equation are used as starting values for the global fit.

For the 5 µM Aurora A/Gleevec complex, the release of the drug was recorded after a 11-fold dilution of the complex using the stopped-flow instrument for 0.25 s (excitation at 295 nm, emission cut-off at 320 nm) at 10°C.

### Creoptix WAVE experiments

Double jump, slow-off, and macroscopic $K_D$ experiments of Aurora A with drugs were studied using a Creoptix WAVE instrument (Creoptix AG, Wädenswil, Switzerland) at 25°C. All chemicals were purchased from GE Healthcare, unless otherwise stated. The protocols in the WAVE control software for conditioning of the chip, immobilization of proteins and performing kinetics experiments were followed. In short, the polycarboxylate chip (PCH) was activated by injection of a 1:1 mixture with

final concentrations of 200 mM N-ethyl-N′-(3-dimethylaminopropyl)carbodiimide (EDC) and 50 mM N-hydroxysuccinimide (NHS), followed by streptavidin immobilization (50 μg/mL in 10 mM sodium acetate pH 5.0). Unreacted sites on the chip were blocked with 1 M ethanolamine pH 8.0. For all activation, immobilization and passivation steps 0.2x HBS-EP was used as running buffer with a flow-rate of 10 μL/min and an injection duration of 420 s on both channels 1 and 2.

Biotinylated T288V variant that mimics dephosphorylated Aurora A was used for experiments performed on the Creoptix WAVE instrument. The activity of T288V with substrate Lats2, the macroscopic $K_D$ and slow-off rate of Danusertib were the same as wild-type (data not shown). Biotinylated T288V Aurora A (70 μg/mL) was immobilized on the PCH-streptavidin chip with 10 μL/min injection and 15 s injection duration over channel 1 only (channel 2 was used as reference channel). All experiments were run in 50 mM HEPES, 50 mM NaCl, 20 mM $MgCl_2$, 5 mM TCEP, 0.03 mg/mL BSA, 0.005% Tween-20, pH 7.30 as running buffer. Binding experiments were evaluated over a range of Danusertib (0.13–66.67 nM), AT9283 (0.03–64.8 nM), and Gleevec (0.37–40 μM) concentrations. Gleevec binding experiments contained 5% DMSO in the running buffer (see above) to enhance Gleevec's solubility. Double-jump experiments of Aurora A/drugs were performed by injecting 1 μM Danusertib or AT9283 with 0.2, 0.4, 0.8, and 2 s injection duration for Danusertib and 1 and 3 s injection duration for AT9283 followed by a 60 s dissociation duration per injection. The slow-off experiments were performed by injecting 5 μM Danusertib or AT9283 with 5–10 s injection duration (to fully saturate Aurora A) followed by a 180 s injection of buffer to remove the excess drug and the dissociation was measured for a duration of 10800 s.

### Spectrofluorometer experiments

The spectrofluorometer FluoroMax-4 (Horiba Scientific) with temperature controller was used to study the slow-off rate of Aurora A with Danusertib at 25°C. For this experiment, a solution containing 30 nM Aurora A and 30 nM Danusertib was pre-incubated for an hour, before diluting 30-fold into degassed buffer (ratio 1:30). A significant decrease in the fluorescence intensity of Aurora A (excitation at 295 nm, emission at 340 nm) can be seen due to the Danusertib release. The fluorescence signal was recorded every 160 s for a duration of six hours using the photobleaching minimization option that will close the shutter after each acquisition. A control experiment was performed, using the same experimental conditions, but without drug in order to account for photobleaching.

## Overall dissociation constant calculated from intrinsic rate constants

In the following equations, $K_1$, $K_2$, $K_3$ and $K_4$ are equal to:

$$K_1 = \frac{k_{-1}}{k_1}$$
$$K_2 = \frac{k_{-2}}{k_2} = \frac{k_{off}}{k_{on}}$$
$$K_3 = \frac{k_{-3}}{k_3}$$
$$K_4 = \frac{k_{-4}}{k_4}$$

Conformational selection followed by inhibitor binding:

$$\mathrm{E_{in}} \underset{k_{-1}}{\overset{k_1}{\rightleftharpoons}} \mathrm{E_{out}} + \mathrm{I} \underset{k_{off}}{\overset{k_{on}}{\rightleftharpoons}} \mathrm{E_{out} \cdot I}$$
$$\quad K_1 \qquad\qquad K_2 \qquad\quad K_D = (K_1 + 1) * K_2$$

(2)

Inhibitor binding followed by an induced-fit step:

$$\mathrm{E_{out}} + \mathrm{I} \underset{k_{off}}{\overset{k_{on}}{\rightleftharpoons}} \mathrm{E_{out} \cdot I} \underset{k_{-3}}{\overset{k_3}{\rightleftharpoons}} \mathrm{E_{out}^* \cdot I}$$
$$\qquad K_2 \qquad\qquad K_3 \qquad\quad K_D = \frac{K_2 * K_3}{(K_3 + 1)}$$

(3)

Conformational selection followed by inhibitor binding and an induced-fit step:

$$E_{in} \underset{k_{-1}}{\overset{k_1}{\rightleftharpoons}} E_{out} + I \underset{k_{off}}{\overset{k_{on}}{\rightleftharpoons}} E_{out} \cdot I \underset{k_{-3}}{\overset{k_3}{\rightleftharpoons}} E_{out}^* \cdot I$$

$$K_1 \qquad K_2 \qquad K_3 \qquad K_D = \frac{(K_1+1)*K_2*K_3}{K_3+1}$$

(4)

Conformational selection mechanism, followed by inhibitor binding to both DFG-*in* and -*out* state, but an induced-fit step only occurs in the DFG-*in* state:

$$E_{out} \underset{k_{-1}}{\overset{k_1}{\rightleftharpoons}} E_{in} + I \underset{k_{off}}{\overset{k_{on}}{\rightleftharpoons}} E_{in} \cdot I \underset{k_{-3}}{\overset{k_3}{\rightleftharpoons}} E_{in}^* \cdot I$$

$$K_4 \Big\updownarrow \quad K_1 \qquad K_2 \qquad K_3 \qquad K_D = \frac{(K_1+1)*K_2*K_3*K_4}{K_1*K_2*K_3+K_3*K_4+K_4}$$

$$E_{out} \cdot I$$

(5)

The uncertainties in the calculated dissociation constant parameter using the equations above are obtained using standard error propagation.

## Aurora A binding to mant-ATP

FRET using intrinsic tryptophan fluorescence is used to monitor mant-ATP (obtained from Jena Bioscience) binding kinetics to Aurora A at 10°C. In the binding experiment or $k_{on}$, increasing concentration of mant-ATP were quickly mixed to 0.5 µM Aurora A (ratio 1:10, excitation at 295 nm, emission cut-off at 395 nm). In the experiment to measure the release of mant-ATP or $k_{off}$, 10 µM/10 µM Aurora A/mant-ATP complex was diluted with buffer (ratio 1:10). A significant decrease in the fluorescence intensity of Aurora A (excitation at 295 nm, emission cut-off at 395 nm) can be seen due to the mant-ATP release.

## Macroscopic dissociation constant experiments

Fluorescence titration experiments were measured using FluoroMax-4 spectrofluorometer (Horiba Scientific). Increasing amounts of Aurora A/Danusertib complex (4 nM Aurora A and 150 nM Danusertib) or Aurora A/mant-ATP (1 µM Aurora A and 2 mM mant-ATP) were titrated into an Aurora A solution (4 nM and 1 µM Aurora A for experiments with Danusertib and mant-ATP, respectively). To measure Danusertib affinity, the excitation wavelength was 295 nm (5 nm bandwidth) and emission spectra were recorded from 310 to 450 nm (20 nm bandwidth) in increments of 2 nm and the temperature was maintained at 25°C. For the mant-ATP experiment, the dissociation constant was measured at 10°C using fluorescence energy transfer from tryptophan residues in Aurora A to mant-ATP by setting the excitation wavelength to 290 nm (5 nm bandwidth) and collecting the emission intensity from 310 to 550 nm (5 nm bandwidth) in increments of 2 nm. A control experiment in the absence of Aurora A was performed using the same experimental settings and used to correct for the mant-ATP interference. In all experiments, a 5 min equilibration time was used after each addition of Aurora A/Danusertib complex or Aurora A/mant-ATP complex.

The fluorescence intensity at 368 nm versus Danusertib concentration or the change in fluorescence at 450 nm ($\Delta F^{450}$) versus mant-ATP concentration was fitted to *Equation 6* using Levenberg-Marquardt nonlinear fitting algorithm included in KaleidaGraph to obtain the $K_D$.

$$F = F_0 + A \cdot \frac{[I] + [E_t] + K_D - \sqrt{([I] + [E_t] + K_D)^2 - 4 \cdot [E_t] \cdot [I]}}{2 \cdot [E_t]}$$

(6)

$F$ and $F_0$ are the fluorescence and initial fluorescence intensities, respectively. [I] and [$E_t$] are the total concentration of the drug or mant-ATP and the Aurora A, respectively.

## Acknowledgements

We thank C. Sassetti (University of Massachusetts Medical School, Worcester) for the use of the ESI-Q-TOF instrument, and the Advanced Light Source (ALS), Berkeley, CA, USA, for access to beamlines BL8.2.1. and BL8.2.2. The Berkeley Center for Structural Biology is supported in part by the National Institutes of Health, National Institute of General Medical Sciences, and the HHMI. The ALS

is supported by the Director, Office of Science, Office of Basic Energy Sciences, of the U.S. Department of Energy under contract DE-AC02-05CH11231. We thank Shohei and Akiko Koide (New York University) for the plasmid of the monody used here. This work was supported by the Howard Hughes Medical Institute (HHMI); the Office of Basic Energy Sciences, Catalysis Science Program, U. S. Dept. of Energy (award DE-FG02-05ER15699); and the NIH (grant GM100966-01). R.O. was a HHMI Fellow of the Damon Runyon Cancer Research Foundation (DRG-2114–12).

## Additional information

### Funding

| Funder | Grant reference number | Author |
|---|---|---|
| Howard Hughes Medical Institute | | Dorothee Kern |
| National Institutes of Health | GM100966-01 | Dorothee Kern |
| U.S. Department of Energy | DE-FG02-05ER15699 | Dorothee Kern |
| Damon Runyon Cancer Research Foundation | DRG-2114-12 | Renee Otten |

The funders had no role in study design, data collection and interpretation, or the decision to submit the work for publication.

### Author contributions

Warintra Pitsawong, Conceptualization, Formal analysis, Investigation, Visualization, Writing—original draft, Writing—review and editing; Vanessa Buosi, Conceptualization, Formal analysis, Investigation, Visualization, Writing—original draft; Renee Otten, Conceptualization, Formal analysis, Funding acquisition, Investigation, Visualization, Writing—original draft, Writing—review and editing; Roman V Agafonov, Conceptualization, Formal analysis, Investigation, Visualization, Writing—review and editing; Adelajda Zorba, Gunther Kern, Conceptualization, Formal analysis, Investigation, Writing—review and editing; Nadja Kern, Xavier Meniche, Formal analysis, Investigation; Steffen Kutter, Ricardo AP Pádua, Formal analysis, Investigation, Writing—review and editing; Dorothee Kern, Conceptualization, Supervision, Funding acquisition, Writing—original draft, Project administration, Writing—review and editing

### Author ORCIDs

Warintra Pitsawong http://orcid.org/0000-0001-5438-1783
Renee Otten http://orcid.org/0000-0001-7342-6131
Adelajda Zorba http://orcid.org/0000-0002-4452-8419
Nadja Kern http://orcid.org/0000-0002-1313-5890
Steffen Kutter http://orcid.org/0000-0002-0349-8161
Ricardo AP Pádua http://orcid.org/0000-0001-9719-3440
Dorothee Kern http://orcid.org/0000-0002-7631-8328

### Decision letter and Author response

Decision letter https://doi.org/10.7554/eLife.36656.039
Author response https://doi.org/10.7554/eLife.36656.040

## Additional files

### Supplementary files

• Transparent reporting form
DOI: https://doi.org/10.7554/eLife.36656.020

### Data availability

Diffraction data have been deposited in PDB under the accession codes 6CPE, 6CPF, 6CPG.

The following datasets were generated:

| Author(s) | Year | Dataset title | Dataset URL | Database, license, and accessibility information |
|---|---|---|---|---|
| Otten R, Kutter S, Buosi V, Padua RAP, Kern D | 2018 | Structure of apo, dephosphorylated Aurora A (122-403) in an active conformation | https://www.rcsb.org/pdb/search/structid-Search.do?structureId=6CPE | Publicly available at RCSB Protein Data Bank (accession no: 6CPE) |
| Otten R, Zorba A, Padua RAP, Kern D | 2018 | Structure of dephosphorylated Aurora A (122-403) bound to AMPPCP in an active conformation | https://www.rcsb.org/pdb/search/structid-Search.do?structureId=6CPF | Publicly available at RCSB Protein Data Bank (accession no: 6CPF) |
| Otten R, Kutter S, Zorba A, Padua RAP, Kern D | 2018 | Structure of dephosphorylated Aurora A (122-403) in complex with inhibiting monobody and AT9283 in an inactive conformation | https://www.rcsb.org/pdb/search/structid-Search.do?structureId=6CPG | Publicly available at RCSB Protein Data Bank (accession no: 6CPG) |

The following previously published datasets were used:

| Author(s) | Year | Dataset title | Dataset URL | Database, license, and accessibility information |
|---|---|---|---|---|
| Zorba A, Kutter S, Cho YJ, Kern D | 2014 | Structure of dephosphorylated Aurora A (122-403) bound to AMPPCP | https://www.rcsb.org/pdb/search/structid-Search.do?structureId=4C3R | Publicly available at RCSB Protein Data Bank (accession no: 4C3R) |
| Cheetham GMT, Knegtel RMA, Coll JT, Renwick SB, Swenson L, Weber P, Lippke JA, Austen DA | 2003 | CRYSTAL STRUCTURE OF AURORA-2, AN ONCOGENIC SERINE-THREONINE KINASE | https://www.rcsb.org/pdb/search/structid-Search.do?structureId=1MUO | Publicly available at RCSB Protein Data Bank (accession no: 1MUO) |
| Bayliss R, Conti E | 2003 | Structure of Human Aurora-A 122-403 phosphorylated on Thr287, Thr288 | https://www.rcsb.org/pdb/search/structid-Search.do?structureId=1OL7 | Publicly available at RCSB Protein Data Bank (accession no: 1OL7) |
| Howard S, Berdini V, Boulstridge JA, Carr MG, Cross DM, Curry J, Devine LA, Early TR, Fazal L, Gill AL, Heathcote M, Maman S, Matthews JE, McMenamin RL, Navarro EF, O'Brien MA, O'Reilly M, Rees DC, Reule M, Tisi D, Williams G, Vinkovic M, Wyatt PG | 2009 | Structure determination of Aurora Kinase in complex with inhibitor | https://www.rcsb.org/pdb/search/structid-Search.do?structureId=2W1G | Publicly available at RCSB Protein Data Bank (accession no: 2W1G) |
| Cameron AD, Izzo G, Storici P, Rusconi L, Fancelli D, Varasi M, Berta D, Bindi S, Forte B, Severino D, Tonani R, Vianello P | 2006 | Structure of Aurora-2 in complex with PHA-739358 | https://www.rcsb.org/pdb/search/structid-Search.do?structureId=2J50 | Publicly available at RCSB Protein Data Bank (accession no: 2J50) |

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
