## [Decision Letter]

Thank you for submitting your article "Dynamics of human protein kinases linked to drug selectivity" for consideration by *eLife*. Your article has been reviewed by three peer reviewers, and the evaluation has been overseen by Philip Cole as the Senior and Reviewing Editor. The following individual involved in review of your submission has agreed to reveal his identity: John Kuriyan (Reviewer #3).

The reviewers have discussed the reviews with one another and the Reviewing Editor has drafted this decision to help you prepare a revised submission.

This manuscript by the Kern group presents an important advance in our understanding of why different ATP-competitive kinase inhibitors have dramatically different binding properties for the same kinase. The authors use an elegant combination of kinetic measurements using intrinsic tryptophan fluorescence, NMR, and X-ray crystallography, to identify discrete conformational states of the Ser/Thr kinase Aurora A, and to correlate the adoption of these states with drug binding. While the flipping of the DFG motif in the activation loop of kinases is often noted as a commonly-observed conformational change that impacts drug binding, here, the authors identify additional conformational changes, on top of the DFG flip, that impact the fast or slow off-rates of different drugs for Aurora A. The nature of these conformational changes is not defined in the paper, but the paper makes an important contribution by establishing these additional conformational changes are important for Aurora, as well as for the cytoplasmic tyrosine kinases for which the Kern group first established this property.

Collectively, these experiments shed light on the nuanced interplay of inhibitor binding and kinase dynamics, and provide an experimental approach to dissect kinetic mechanisms of drug binding that could play an important role in future kinase inhibitor design strategies. The main conclusion is that the dynamical response to inhibitor binding that was established earlier (by this group) for the tyrosine kinases Src and Abl also applies to Aurora, a Ser/Thr kinase. As such, this work merits publication in *eLife*, after revision. While revising the manuscript, the authors should pay careful attention to placing this work in proper context within what is already known and appreciated about how kinases respond to inhibitors. Also, the work would be strengthened by being clearer about what is actually learned, rather than overstating the conclusions. This work advances the field in an important way, and would be valued all the more if it represented past insights more appropriately. In this regard, the title of the paper should be made more specific, and not imply that a broad cross-section of kinases have been studied, or that a single mechanism has been uniquely defined.

Essential revisions:

1) The way the paper is written the reader may get the mistaken impression that the DFG conformational change is irrelevant (or less important) for drug selectivity.

Indeed, although previous work from the Kern group established that an induced conformational change after Gleevec binding to Abl, but not to Src, contributes to differential inhibition of these two kinases, it remains true that Gleevec can only bind to either of these kinases in the DFG-out state. Prior to the original observation of the DFG flip in Abl kinase, by the Kuriyan group, the DFG-out state was not known to be an accessible conformational state of protein kinases, let alone one that could bind a drug.

Related to this point, the authors aim to describe a new, generalizable paradigm for kinase-inhibitor interactions, one that decreases the importance of the DFG-flip. While this is intriguing, it awaits further analysis of the conformational landscape of other kinases, in the presence and absence of inhibitors. The authors should consider rephrasing sections so that they do not downplay the significance of the DFG-out conformation in drug binding. In other words, nothing that has followed since the original crystal structure of Abl bound to Gleevec (Schindler et al.) has overturned the conclusion drawn by the Kuriyan group that Gleevec specifically recognizes a DFG-flipped conformation of the kinase. What is now appreciated is that the DFG-flip is not the only requirement for drug binding, and many (if not all) kinases can undergo this conformational change. The way this paper is written, this is not obvious. Also, at points where the "seminal" observation of Gleevec selectivity is described, the authors cite much more recent papers, rather than the original discoveries. The work from the Kuriyan group is cited, but in a throw-away sentence in the middle of the paper.

2) The authors somewhat mischaracterize our general understanding of the role of activation loop phosphorylation. While it is true that many seminal studies have established a correlation between activation loop phosphorylation and the adoption of an active, rather inactive configuration, several studies have actually suggested that this is a conformational selection, rather than induced fit, phenomenon. Indeed, some protein kinases lacking activation loop phosphorylation have been crystalized in an active conformation, and molecular dynamics calculations suggest that this the active state is accessible to the protein kinases in the absence of activation loop phosphorylation, albeit at a higher energetic cost (see work by Benoit Roux, and also Vijay Pande). Most recently, the work by Ruff et al. on Arora A (2018), the same kinase studied in this report, has convincingly shown that activation loop phosphorylation does not preclude access to an inactive conformation in this kinase.

3) The issue of an unphosphorylated kinase domain adopting different conformations in crystals is presented as surprising, but this is well appreciated. These points were also made in Levinson et al. (2006) (not cited here) where the authors present "crystal structures of the Abl kinase domain bound to substrate-mimicking ATP analog-peptide conjugates that represent four distinct conformations of the unphosphorylated kinase domain". The idea that two different conformations can be found in the same crystal is also not new. Crystallographically, this was noted in Seeliger et al., 2007, who report a crystal structure in which two molecules of Src are found, one bound to Gleevec, and the other in an active conformation. Given these points, the authors could consider rewording parts of the first two paragraphs of the Results section so as not to suggest that their observation of an active-like unphosphorylated kinase is unexpected.

4) The authors introduce the paper by pointing out that it remains a challenge to understand why certain drugs are selective for individual or few kinases out of the whole kinome (~500 related enzymes). They present the work in this manuscript with the intention of solving this problem. The work presented here, however, only explains how different inhibitors interact with one individual kinase, not why the same inhibitor might have differential binding properties for different kinases. This crossing of concepts is misleading to the readers.

5) The fact that kinases can undergo a conformational change that results in remarkably slow off-rates for inhibitors has been appreciated from work such as the binding of Tykerb to EGFR (Wood et al., 2004). This point should be made in the revised paper, and the appropriate papers cited.

6) The concept that the DFG flip is not the sole determining factor in determining the binding of Gleevec to Abl vs Src was first established by determining the structures of high-affinity compounds bound to Src in a DFG-flipped conformation by Seeliger et al. 2009, Cancer Research and Dar et al., 2008, Chemical Biology. Neither paper is cited in the present paper, which gives readers the misleading impression that the field was not aware of this fact prior to the Kern contributions.

7) As noted earlier, a related paper was just published in *eLife* by Ruff et al. (2018), examining the impact of activation loop phosphorylation on kinase dynamics and the DFG-flip. In that report, the investigators conclude that Aurora A can sample the DFG-in and DFG-out states with and without activation loop phosphorylation, and that another drug, SNS-314 induces further conformational changes upon binding the DFG-in state of Arora A. These results are pertinent to several points in the manuscript reviewed here and should be discussed.

8) The paper relies heavily on elegant kinetic studies using stopped-flow fluorescence measurements, and these measurements yield multi-step kinetic models for drug binding. These models are crucial to this study, but multi-step reactions are hard to intuitively understand. It would be worthwhile for the authors to model and graphically depict the outcomes of several scenarios varying one or two parameters in their kinetic models at a time, to show why, for example, conformational selection between the DFG-in and DFG-out states does not substantially contribute to Danusertib binding, whereas the induced fit step does. Similar analyses could be done for the other inhibitors as well and would greatly facilitate the readers' understanding of these important concepts.

9) It might be interesting to examine the kinetic mechanism of one drug against a few different kinases. The authors already did this in their previous paper, comparing Gleevec binding with Src and Abl, and now with Aurora A in this manuscript. To infer that their observed induced fit phenomena underly a general principle of drug binding, perhaps the authors could compare Danusertib binding kinetics for Aurora A with a few other kinases. If this is not feasible, then the revised manuscript should state that the generality of the principles awaits further study.

---

## [Author Response]

[…] While revising the manuscript, the authors should pay careful attention to placing this work in proper context within what is already known and appreciated about how kinases respond to inhibitors. Also, the work would be strengthened by being clearer about what is actually learned, rather than overstating the conclusions. This work advances the field in an important way, and would be valued all the more if it represented past insights more appropriately. In this regard, the title of the paper should be made more specific, and not imply that a broad cross-section of kinases have been studied, or that a single mechanism has been uniquely defined.

As described in more detail below, we have edited manuscript to be more nuanced and attempted to acknowledge the vast amount of literature in the protein kinase field, following the suggestions by the reviewers. We made the title more specific: “Dynamics of human protein kinase Aurora A linked to drug selectivity”.

Essential revisions:1) The way the paper is written the reader may get the mistaken impression that the DFG conformational change is irrelevant (or less important) for drug selectivity.Indeed, although previous work from the Kern group established that an induced conformational change after Gleevec binding to Abl, but not to Src, contributes to differential inhibition of these two kinases, it remains true that Gleevec can only bind to either of these kinases in the DFG-out state. Prior to the original observation of the DFG flip in Abl kinase, by the Kuriyan group, the DFG-out state was not known to be an accessible conformational state of protein kinases, let alone one that could bind a drug.Related to this point, the authors aim to describe a new, generalizable paradigm for kinase-inhibitor interactions, one that decreases the importance of the DFG-flip. While this is intriguing, it awaits further analysis of the conformational landscape of other kinases, in the presence and absence of inhibitors. The authors should consider rephrasing sections so that they do not downplay the significance of the DFG-out conformation in drug binding. In other words, nothing that has followed since the original crystal structure of Abl bound to Gleevec (Schindler et al.) has overturned the conclusion drawn by the Kuriyan group that Gleevec specifically recognizes a DFG-flipped conformation of the kinase. What is now appreciated is that the DFG-flip is not the only requirement for drug binding, and many (if not all) kinases can undergo this conformational change. The way this paper is written, this is not obvious. Also, at points where the "seminal" observation of Gleevec selectivity is described, the authors cite much more recent papers, rather than the original discoveries. The work from the Kuriyan group is cited, but in a throw-away sentence in the middle of the paper.

The reviewers are correct that the DFG-state is important for whether a drug can or cannot bind to the kinase due to steric clash. The seminal work by the Kuriyan lab on Gleevec binding to Abl and Src, and other examples by various research groups, have provided unequivocal evidence for this fact. We certainly did not intend to downplay any of these contributions nor the significance of the orientation of DFG-motif in the physical drug-binding step. We had elected to cite reviews and several of the more recent papers, but the reviewers are correct that we should include more of the original research papers. We have now added these references. A conformational-selection mechanism alone has however proved insufficient to explain drug affinity and selectivity, and in fact, will by definition weaken the overall affinity. We show here (and in our earlier work on Gleevec) that conformational dynamics after drug binding plays the crucial role in understanding overall selectivity and affinity. We realize that our original wording created some misunderstanding and have edited the manuscript to reflect this sentiment more clearly; in the introduction it now states:

“Seminal work by the Kuriyan lab demonstrated that Gleevec can only bind to an inactive DFG (for Asp-Phe-Gly) loop conformation in the “out-conformation” due to steric clash of the active, DFG-in conformation (Nagar et al., 2002; Schindler et al., 2000; Seeliger et al., 2007). […] Recent quantitative binding kinetics combined with ancestral sequence reconstruction put forward a mechanism where an induced-fit step after drug binding is the key determinant for Gleevec’s selectivity (Agafonov, Wilson, Otten, Buosi, and Kern, 2014; Wilson et al., 2015), and fully recapitulates the binding affinities.”

2) The authors somewhat mischaracterize our general understanding of the role of activation loop phosphorylation. While it is true that many seminal studies have established a correlation between activation loop phosphorylation and the adoption of an active, rather inactive configuration, several studies have actually suggested that this is a conformational selection, rather than induced fit, phenomenon. Indeed, some protein kinases lacking activation loop phosphorylation have been crystalized in an active conformation, and molecular dynamics calculations suggest that this the active state is accessible to the protein kinases in the absence of activation loop phosphorylation, albeit at a higher energetic cost (see work by Benoit Roux, and also Vijay Pande). Most recently, the work by Ruff et al. on Arora A (2018), the same kinase studied in this report, has convincingly shown that activation loop phosphorylation does not preclude access to an inactive conformation in this kinase.

We certainly did not intend to mischaracterize the understanding of the role of activation loop phosphorylation. We describe the general notion from X-ray crystallography data that unphosphorylated, apo Aurora A kinase is mostly found in an inactive conformation, whereas an active conformation is observed when the activation loop is phosphorylated or upon TPX2 binding. Whether the interconversion from an inactive to the active conformation happens before (i.e., conformational selection) or after (i.e., induced fit) phosphorylation of the kinase is a question of flux. This can only be addressed by measuring the kinetics of Aurora A phosphorylation in concert with its conformational interconversion between the inactive and active conformation. In other words, a kinetic flux analysis for protein phosphorylation would be required, similarly to the kinetic flux analysis performed here for the inhibitors. This topic is outside of the scope of this manuscript and not addressed here, and, to the best of our knowledge, such kinetics analysis has not been reported thus far.

We fully agree with the reviewers that in the absence of phosphorylation the protein can sample an active conformation. In fact, the data in our manuscript on the dephosphorylated kinase provides three lines of evidence for this: (1) structures of apo and AMPPCP-bound Aurora A crystallized in the active conformation, (2) stopped-flow fluorescence data showing the presence of a conformational-selection mechanism, and (3) NMR spectra indicating exchange between different states. We have expanded the section describing the computational efforts and included more references:

“Owing to the reported importance of the DFG flip for activity, regulation and drug design, there have been extensive efforts to characterize this conformational equilibrium by computation (Badrinarayan and Sastry, 2014; Barakat et al., 2013; Meng, Lin, and Roux, 2015; Meng, Pond, and Roux, 2017; Sarvagalla and Coumar, 2015; Shukla, Meng, Roux, and Pande, 2014). The general notion of these computational studies is that in the absence of phosphorylation the inactive form of the kinase is most favored, in agreement with experimental evidence. Nevertheless, short-lived excursions to the active state are observed.”

More recently, experiments on Aurora A kinase described by Cyphers et al. (2017), Gilburt et al. (2017) and Ruff et al. (2018) report on the position of the activation loop of (un)phosphorylated kinase in the absence/presence of various small molecules. For monitoring the DFG-loop, an IR probe was chemically incorporated into a position that hydrogen-bonds to the DFG-loop in the wild-type protein, resulting in a large reduction in kinase activity. We therefore opt for caution in the interpretation of these data in respect of the DFG-loop conformation for the wild-type protein. Our results are consistent with their observations in that Aurora A samples several activation loop conformations in its unphosphorylated form and that binding of nucleotides merely affects the equilibrium constant (Cyphers et al., 2017). Since our manuscript only pertains to dephosphorylated Aurora A, we have no data to compare to the experiments described by Gilburt et al. and Ruff et al.that report on phosphorylated Aurora A.We have extended the first paragraph of the Results section to include these more recent papers:

“X-ray structures, however, provide merely static snapshots of possible kinase conformations that do not necessarily reflect the situation in solution. In fact, recent experimental data postulate that phosphorylation of Aurora A does not “lock” the kinase in the active conformation, and that the activation-loop still exhibits conformational dynamics (Gilburt et al., 2017; Ruff et al., 2018). On the other hand, X-ray crystallography provides high-resolution structural data that cannot readily be obtained from FRET or EPR and IR spectroscopy.”

3) The issue of an unphosphorylated kinase domain adopting different conformations in crystals is presented as surprising, but this is well appreciated. These points were also made in Levinson et al. (2006) (not cited here) where the authors present "crystal structures of the Abl kinase domain bound to substrate-mimicking ATP analog-peptide conjugates that represent four distinct conformations of the unphosphorylated kinase domain". The idea that two different conformations can be found in the same crystal is also not new. Crystallographically, this was noted in Seeliger et al., 2007, who report a crystal structure in which two molecules of Src are found, one bound to Gleevec, and the other in an active conformation. Given these points, the authors could consider rewording parts of the first two paragraphs of the Results section so as not to suggest that their observation of an active-like unphosphorylated kinase is unexpected.

We fully agree with the reviewers that our choice of words could be misconstrued. We have reworded the paragraph owing to the well-known fact that different conformations out of the structural ensemble have been captured by X-ray crystallography on a number of systems before. We thank the reviewers for pointing this out and we have included additional references.

“Our results are consistent with other crystallographic studies on wild-type, dephosphorylated Aurora A in its apo or nucleotide bound state, where the kinase was also found in the active conformation (Gustafson et al., 2014; Janecek et al., 2016; Nowakowski et al., 2002).”

We further clarify that in our case we have “Two crystals from the same crystallization well”, not two different conformations within one crystal, or two molecules in the asymmetric unit that are different in structure.

4) The authors introduce the paper by pointing out that it remains a challenge to understand why certain drugs are selective for individual or few kinases out of the whole kinome (~500 related enzymes). They present the work in this manuscript with the intention of solving this problem. The work presented here, however, only explains how different inhibitors interact with one individual kinase, not why the same inhibitor might have differential binding properties for different kinases. This crossing of concepts is misleading to the readers.

The reviewers are correct that the work presented here explains how different inhibitors interact with Aurora A kinase and not the whole kinome. We note, however, that the inhibitor Gleevec was used both for Aurora A and in our earlier work on Src, Abl, and their evolutionary intermediates (Agafonov et al., 2014; Wilson et al., 2015). We find it interesting that the physical binding step of Gleevec to these different kinases is virtually unaltered and that the major contribution for a tight-binder is due to the induced-fit step after binding. The critical role of the dynamics in the drug-bound state holds true when looking at the binding of different small molecules to the Aurora A kinase. We fully agree that the presented results raise the interest to extend our knowledge of the interaction of other inhibitors with a broader cross-section of the kinome (see also point 9, added to the Discussion), which we hope to provoke with our work. We thank the reviewers for bringing up our misleading wording. We have changed the text accordingly and made the comparison for Gleevec binding to the different protein kinases clearer in the Results section.

5) The fact that kinases can undergo a conformational change that results in remarkably slow off-rates for inhibitors has been appreciated from work such as the binding of Tykerb to EGFR (Wood et al., 2004). This point should be made in the revised paper, and the appropriate papers cited.

While we referred in the Discussion section to literature reviews on the “drug-target residence time”, we can certainly make this more specific and describe some of the earlier work on kinases. We thank the reviewers for pointing this out and have included this in the revised manuscript in the Results section when describing the danusertib experiments:

“Earlier examples of protein kinases that also show remarkable slow off-rates, presumably caused by conformational changes, include the epidermal growth factor receptors (Berezov, Zhang, Greene, and Murali, 2001; Wood et al., 2004) and CDK8 (Schneider, Bottcher, Huber, Maskos, and Neumann, 2013) amongst others (Willemsen-Seegers et al., 2017). To the best of our knowledge, we present here for the first time a detailed stopped-flow kinetics analysis for Aurora A that unequivocally shows the slow off-rate is caused by the conformational change within the drug-bound state, and not the dissociation step.”

6) The concept that the DFG flip is not the sole determining factor in determining the binding of Gleevec to Abl vs Src was first established by determining the structures of high-affinity compounds bound to Src in a DFG-flipped conformation by Seeliger et al. 2009, Cancer Research and Dar et al., 2008, Chemical Biology. Neither paper is cited in the present paper, which gives readers the misleading impression that the field was not aware of this fact prior to the Kern contributions.

It was indeed recognized earlier that the DFG-flip alone could not rationalize the binding of Gleevec to Abl/Src, and other hypotheses were put forward. We had no intent to represent this otherwise and, for clarification, we have modified the introduction to reflect this (see under point 1). Here, we provide now the relative contribution of the individual steps in the coupled equilibria for the binding of inhibitors to Aurora A via quantitative kinetic experiments.

7) As noted earlier, a related paper was just published in eLife by Ruff et al. (2018), examining the impact of activation loop phosphorylation on kinase dynamics and the DFG-flip. In that report, the investigators conclude that Aurora A can sample the DFG-in and DFG-out states with and without activation loop phosphorylation, and that another drug, SNS-314 induces further conformational changes upon binding the DFG-in state of Arora A. These results are pertinent to several points in the manuscript reviewed here and should be discussed.

See our answer to point 2.

8) The paper relies heavily on elegant kinetic studies using stopped-flow fluorescence measurements, and these measurements yield multi-step kinetic models for drug binding. These models are crucial to this study, but multi-step reactions are hard to intuitively understand. It would be worthwhile for the authors to model and graphically depict the outcomes of several scenarios varying one or two parameters in their kinetic models at a time, to show why, for example, conformational selection between the DFG-in and DFG-out states does not substantially contribute to Danusertib binding, whereas the induced fit step does. Similar analyses could be done for the other inhibitors as well and would greatly facilitate the readers' understanding of these important concepts.

We agree with the reviewers that the fine details of multi-step reactions and coupled-equilibria are often not intuitive to understand. The contribution of a conformational-selection and induced-fit step on the overall dissociation constant can be appreciated in a quite straightforward manner from Equation 4 (i.e., thermodynamics of coupled equilibria) in the manuscript:

1) a conformational selection mechanism weakens the overall affinity by the amount of sampling of the binding-incompetent state

2) an induced-fit step tightens the overall affinity by the amount of “right-shifting” the equilibrium to the E*:D state.

We demonstrate this graphically in Author response image 1 for the binding of danusertib to Aurora A, as suggested by the reviewers. In panels A and B, we change the equilibrium constant for the conformational selection mechanism (K_1_) or induced-fit step (K_3_) by two orders of magnitude in both directions, starting from unity. For comparison, the K_D_ value is shown when only the physical drug-binding step would be present (the value for K_2_, 14 𝜇M, is taken from our stopped-flow experiments, see Figure 5). In panel C we show the effect of changing the same equilibrium constants for the combined, four-state model (see Figure 5I, Equation 4). Both K_1_ and K_3_ are changed by two orders of magnitude around the experimentally determined values of 0.67 and 3.7·10^-5^, respectively, and the overall K_D_ is plotted on the y-axis. If the reviewers feel that this would be useful to include as a figure supplement, we would be happy to do so. Alternatively, should we move Equation 4 into the main text?

9) It might be interesting to examine the kinetic mechanism of one drug against a few different kinases. The authors already did this in their previous paper, comparing Gleevec binding with Src and Abl, and now with Aurora A in this manuscript. To infer that their observed induced fit phenomena underly a general principle of drug binding, perhaps the authors could compare Danusertib binding kinetics for Aurora A with a few other kinases. If this is not feasible, then the revised manuscript should state that the generality of the principles awaits further study.

Agreed and we revised the manuscript accordingly:

“From our results on Aurora A kinase presented here and earlier data on Tyrosine-kinases (Agafonov et al., 2014; Wilson et al., 2015), we propose that this may be a general mechanism for different kinases and multiple inhibitors, thereby providing a platform for future computational and experimental efforts in rational drug design. Albeit, we note that verification of this proposition requires a larger sampling of small molecules and different protein kinases throughout the kinome.”

We hope that this manuscript will trigger similar studies in many labs working on exciting aspects of many different kinases/inhibitor interactions to test how general this concept is (we speculate it will be quite general).